# From Noise to Intent: Anchoring Generative VLA Policies with Residual Bridges

**Yiming Zhong** [* 1]  **Yaoyu He** [* 1]  **Zemin Yang** [* 1]  **Pengfei Tian** [1]  **Yifan Huang** [1]  **Qingqiu Huang** [2]  **Xinge Zhu** [3]
**Yuexin Ma** [† 1]

## Abstract

Bridging high-level semantic understanding with low-level physical control remains a persistent challenge in embodied intelligence, stemming from the fundamental spatiotemporal scale mismatch between cognition and action. Existing generative VLA policies typically adopt a "Generation-from-Noise" paradigm, which disregards this disparity, leading to representation inefficiency and weak condition alignment during optimization. In this work, we propose **ResVLA**, an architecture that shifts the paradigm to "Refinement-from-Intent." Recognizing that robotic motion naturally decomposes into global intent and local dynamics, ResVLA utilizes spectral analysis to decouple control into a deterministic low-frequency anchor and a stochastic high-frequency residual. By anchoring the generative process on the predicted intent, our model focuses strictly on refining local dynamics via a residual diffusion bridge. Extensive simulation experiments show that ResVLA achieves competitive performance, strong robustness to language and robot embodiment perturbations, and faster convergence than standard generative baselines. It also demonstrates strong performance in real-world robot experiments.

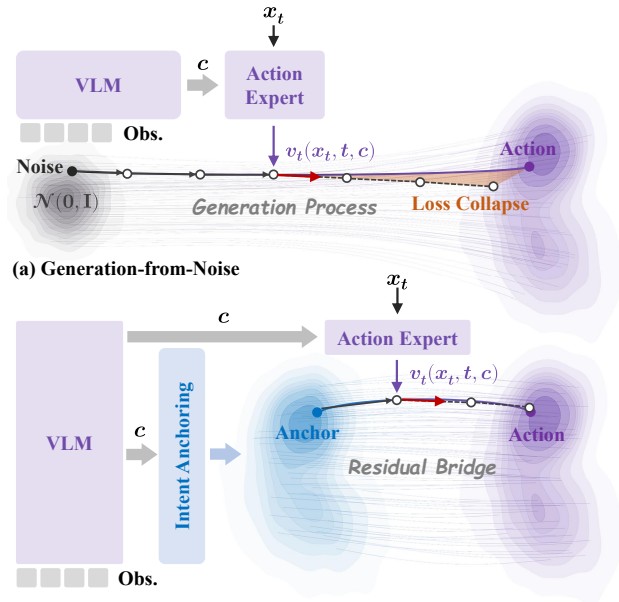

*Figure 1.* Paradigm comparison. (a) Generation-from-Noise initializes from uninformative noise, leading to inefficient transport paths and "Loss Collapse" where trajectories fail to align with semantic instructions. (b) Refinement-from-Intent (Ours) anchors generation on a predicted low-frequency intent. This establishes a short-path "Residual Bridge" that focuses strictly on refining high-frequency dynamics to reach the target action manifold.

## 1. Introduction

*"We build too many walls and not enough bridges."*

— *Isaac Newton*

---

[*]Equal contribution
**Project Page:** https://res-vla.github.io/ResVLA/
**Code:** https://github.com/4DVLab/ResVLA
[1]ShanghaiTech University, Shanghai, China [2]Morphic Robotics, Shenzhen, China [3]The Chinese University of Hong Kong, Hong Kong, China. Correspondence to: Yiming Zhong <zhongym2024@shanghaitech.edu.cn>, Yuexin Ma <mayuexin@shanghaitech.edu.cn>.

*Proceedings of the 43rd International Conference on Machine Learning*, Seoul, South Korea. PMLR 306, 2026. Copyright 2026 by the author(s).

The rapid advancement of Vision-Language-Action (VLA) models has endowed generalist robots with remarkable capabilities in comprehending complex semantic instructions (Brohan et al., 2023b;a; Kim et al., 2024; 2025; Black et al., 2026; Pertsch et al., 2025; Bjorck et al., 2025). However, effectively grounding this high-level semantic understanding into low-level physical manipulation remains a fundamental challenge in embodied intelligence. We posit that this challenge stems from a spatiotemporal scale mismatch between cognition and action. Specifically, the cognitive intent derived from VLMs operates at a macro-temporal scale, prioritizing global trajectory planning and long-horizon geometric consistency (Ahn et al., 2022; Huang et al., 2022). In terms of signal characteristics, this manifests as a low-frequency and deterministic distribution. In contrast, successful physical interaction necessitates precise modulation

at a micro-temporal scale to accommodate contact dynamics, friction, and sensor noise (Hogan, 1984; Lee et al., 2019; Levine et al., 2015). These execution details inherently exhibit a high-frequency and highly stochastic distribution. This disparity suggests that ideal robotic control should not be a process of generation *ex nihilo*, but rather one of **Iterative Refinement**: a process that progressively injects microscopic physical dynamics while strictly adhering to the guidance of the macroscopic semantic structure.

To surmount the limitations of early discrete tokenization approaches in action precision and smoothness (Brohan et al., 2023b;a; Team et al., 2024; Kim et al., 2024), robotic policy research has pivoted towards a continuous generative policy paradigm. Representative architectures, such as $\pi_0$ (Black et al., 2026) and $\pi_{0.5}$ (Intelligence et al., 2025), adopt continuous generative action models based on Flow Matching (Lipman et al., 2022) and diffusion policies (Chi et al., 2025), significantly enhancing physical fidelity and multimodal modeling capabilities. However, while these methods excel at capturing complex physical dynamics, they predominantly adhere to a **"Generation-from-Noise"** paradigm, forcing the model to reconstruct the entire action distribution starting from an uninformative, isotropic Gaussian prior $\mathcal{N}(\mathbf{0}, \mathbf{I})$. This approach disregards the aforementioned essence of refinement, complicating task execution into a problem of conditional distribution modeling from scratch. The cost is primarily twofold: (1) Representation Inefficiency: the model is compelled to expend the velocity field on rediscovering the explicit global intent, which is computationally wasteful; (2) Loss Collapse: as highlighted by recent theoretical analyses (Pan et al., 2025; Dong et al., 2025), the independence between the initial noise source and the task condition renders the optimization prone to ignoring fine-grained language instructions. Consequently, the generated actions, while statistically plausible in physics, often fail to align with the critical semantic intent.

Given the structural flaws of **"Generation-from-Noise"**, we advocate for a return to the essence of control by establishing a new paradigm of **"Refinement-from-Intent"** (Figure 1). Realizing this paradigm requires addressing two core questions: **Where to start?** and **How to refine?** Regarding the first question, we identify that spectral analysis offers a natural perspective for decoupling. Since low-frequency components encapsulate the global geometric structure of the trajectory, they naturally constitute the effective anchor for this refinement process, collapsing uncertain intent into a deterministic prior. Regarding the second question, we introduce the Diffusion Bridge (Bortoli et al., 2023) mechanism to construct the refinement path. Unlike standard diffusion models that perform blind exploration from Gaussian noise, diffusion bridge models explicitly establish a directed connection from a known starting point (i.e., our low-frequency intent) to the target distribution (ground-truth actions). Crucially, because the low-frequency intent is geometrically proximate to the true action, this bridging process no longer requires reconstructing the entire trajectory but only filling in the missing high-frequency residuals. This not only dramatically shortens the generative path in geometry, but also physically aligns with the control intuition of fine-tuning only local dynamics.

We instantiate this theoretical framework as **ResVLA**, a general architecture that leverages spectral analysis to decouple intent anchoring and residual refinement. We extensively evaluated ResVLA across a diverse suite of benchmarks, including LIBERO (Liu et al., 2023a), LIBERO-Plus (Fei et al., 2025), and SimplerEnv (Li et al., 2024b). This evaluation suite covers a broad spectrum of challenges, ranging from long-horizon semantic planning to high-fidelity contact manipulation and cross-embodiment generalization. Experimental results demonstrate that ResVLA achieves strong overall performance across this extensive testing spectrum. Specifically, thanks to the semantic locking effect of the low-frequency anchor, our method exhibits significantly superior stability in long-horizon tasks compared to pure generative baselines such as $\pi_0$. Meanwhile, the short-path characteristic of the residual bridge enables the model to more efficiently master complex contact dynamics. Furthermore, substantial improvements in training convergence speed and inference efficiency further validate that introducing structured physical priors into large models is a critical pathway toward efficient and robust generalist robot control.

Our contributions are summarized as follows:

- We identify source-condition independence as a factor behind training inefficiency and loss collapse in generative VLA policies. To address this, we propose a residual refinement perspective that maximizes the mutual information between the source and the condition.

- We introduce **ResVLA**, which fuses deterministic regression and generative flow matching via spectral orthogonality. By anchoring generation on a condition-dependent low-frequency source, we effectively reconcile the conflict between global semantic alignment and local dynamic fidelity.

- Extensive evaluations demonstrate that our **ResVLA** achieves strong performance. Notably, it exhibits superior robustness in long-horizon and contact-rich tasks while converging faster than denoising baselines, validating the efficiency of our residual bridging paradigm.

## 2. Related Work

### 2.1. Evolution of Vision-Language-Action Models

The early development of generative robotic control was primarily dominated by the discrete autoregressive paradigm.

With RT-2 (Brohan et al., 2023a) and OpenVLA (Kim et al., 2024) as cornerstones, this lineage leveraged the logical reasoning capabilities of LLMs, while subsequent works augmented this architecture through various mechanisms (Zhang et al., 2024; Lu et al., 2025a; Zhao et al., 2025; Cen et al., 2025). However, despite their powerful semantic planning capabilities, the quantization error inherent in discrete tokenization remains an insurmountable barrier for fine-grained physical manipulation. To break through this precision bottleneck, the field has pivoted towards the continuous generative paradigm. Represented by Diffusion Policy (Chi et al., 2025), Octo (Team et al., 2024), and $\pi_0$ (Black et al., 2026), these methods achieved high-fidelity physical control by directly modeling continuous distributions. Subsequent works further refined this direction (Pertsch et al., 2025; Intelligence et al., 2025; Qu et al., 2025; Zheng et al., 2024; Shukor et al., 2025; Wang et al., 2025b). Although the continuous paradigm resolved precision issues, most of these models still follow the **"Generation-from-Noise"** paradigm, compelling them to reconstruct intent from uninformative noise, leading to training inefficiency and potential instruction failure. **ResVLA** aims to rectify this structural deficiency: we reject the notion of generation from scratch, instead utilizing the low-frequency intent determined by the VLM as an anchor and employing the model solely to refine high-frequency physical dynamics via our residual diffusion bridge.

### 2.2. Diffusion Bridges and Optimization Pathology

Diffusion Bridges, such as Schrödinger Bridges (De Bortoli et al., 2021) and I$^2$SB (Liu et al., 2023b), generalize standard diffusion by enabling probabilistic connections between source and target distributions. This flexibility establishes an ideal framework for refining sub-optimal priors (e.g., coarse trajectories) into optimal posteriors, showing promise in image restoration (Wang et al., 2025a) and robotic motion planning (Nguyen et al., 2025). However, in conditional control, source design plays an important role in optimization stability. Recent theoretical analyses (Dong et al., 2025) identify a phenomenon termed **"Loss Collapse"**: when the source distribution is independent of the task condition (e.g., instructions), optimization can suffer from weakened conditioning signals, causing the model to ignore fine-grained conditions. Related to this issue, recent work has also explored condition-aware or informative source design. In diffusion policies, Cocos (Dong et al., 2025) introduces condition-dependent sources to mitigate loss collapse; VITA (Gao et al., 2025b) uses visual latents as the source of flow; CAR-Flow (Chen et al., 2025) reparameterizes source and target distributions in a condition-aware manner; and Prior Does Matter (Ren et al., 2025) and Don't Start from Scratch (Walke et al., 2023) show the benefits of informative priors beyond pure Gaussian initialization. The

core contribution of **ResVLA** lies not in proposing a new bridge algorithm, but in instantiating condition-dependent source construction for VLA control. By constructing a condition-dependent source, we enable stable application of diffusion bridges to VLA control.

### 2.3. Structured Priors in Robotic Control

To construct such a robust source distribution, incorporating structured priors has been a longstanding strategy in robotic control. Residual learning classically employs analytical controllers (e.g., PID) as baselines, learning only residual actions to compensate for dynamic errors (Silver et al., 2018; Johannink et al., 2018). In the generative domain, approaches like Decision Diffuser (Ajay et al., 2023) have also embraced this intuition, demonstrating how trajectory generation can be formulated as an iterative refinement process under constraints. In representation learning, frequency and hierarchical structures offer another form of prior: FAST (Pertsch et al., 2025) utilizes DCT for action compression; FreqPolicy (Zhong et al., 2025a) validates the stability of low-frequency prediction; and H$^3$DP (Lu et al., 2025b) optimizes long-horizon generation via spatiotemporal hierarchy. **ResVLA** revisits these concepts within a generative framework. Unlike traditional residual learning that relies on handcrafted baselines and is distinct from the specific architectural designs of FreqPolicy or H$^3$DP, we leverage the data-driven low-frequency intent as a general structured anchor. This not only provides a semantically grounded starting point for the diffusion bridge but also transforms the generation task into a more optimization-friendly residual refinement problem, achieving dual gains in efficiency and precision.

## 3. Theoretical Formulation: Control via Iterative Refinement

### 3.1. Refinement Dynamics in Control Policies

The core objective of robot learning is to approximate the expert conditional distribution $p_1(\mathbf{x}|\mathbf{c})$. Following the insights from Minimal Iterative Policy (MIP) (Pan et al., 2025), we posit that the efficacy of a policy in high-precision tasks stems not merely from distributional matching, but from its capability for **Iterative Refinement**, the ability to repeatedly project an initial guess back onto the expert manifold $\mathcal{M}$ during inference. We analyze existing control paradigms through this lens.

**Deterministic Regression.** Regression-based methods (e.g., ACT (Zhao et al., 2023)) model the policy as a deterministic mapping $\mathbf{x} = f_\theta(\mathbf{c})$, minimizing the expected risk:

$$\min_\theta \mathbb{E}_{(\mathbf{c},\mathbf{x}^*)\sim\mathcal{D}}\|\mathbf{x}^* - f_\theta(\mathbf{c})\|^2. \tag{1}$$

Mathematically, this collapses the target distribution into a

Dirac delta $p(\mathbf{x}|\mathbf{c}) = \delta(\mathbf{x} - f_\theta(\mathbf{c}))$. While computationally efficient, this formulation represents a single-step projection. As noted in MIP, while this paradigm performs adequately in simple tasks, in high-precision manipulation, the lack of subsequent iterative computation deprives the model of the mechanism to correct initial prediction deviations. Consequently, predicted trajectories are prone to drifting off the expert manifold due to error accumulation, manifesting as the "regression to the mean" phenomenon where high-frequency details are lost.

**Discrete Autoregressive Generation.** To incorporate sequential dependencies, methods like OpenVLA (Kim et al., 2024) discretize the action space into token sequences $\mathbf{x} \approx (s_1, \ldots, s_L)$. The generation process factorizes as: $p(\mathbf{x}|\mathbf{c}) = \prod p(s_i|s_{<i}, \mathbf{c})$. While acting as a sequential subspace refinement, this paradigm suffers from a fundamental precision bottleneck:

**Proposition 3.1** (Irreducible Quantization Noise). *For a uniform quantization $\mathcal{Q}$ with resolution $\Delta$, assuming locally uniform ground-truth $\mathbf{x}^*$, the expected reconstruction MSE is strictly lower-bounded:*

$$\mathbb{E}_{\mathbf{x}^*}\left[\|\mathbf{x}^* - \mathcal{Q}^{-1}(\mathcal{Q}(\mathbf{x}^*))\|^2\right] = \frac{\Delta^2}{12}. \quad (2)$$

This term $\Delta^2/12$ represents a permanent **structural deadband**. This irreducible noise prevents asymptotic zero-error convergence, rendering discrete policies theoretically inadequate for high-precision manipulation.

**Continuous Generative Models.** Continuous methods (e.g., diffusion policy, $\pi_0$) model action generation as a time-reversal stochastic process. The refinement dynamics are governed by the reverse-time SDE:

$$d\mathbf{x}_t = [\mathbf{f}(\mathbf{x}_t, t) - g^2(t)\nabla_{\mathbf{x}_t} \log p_t(\mathbf{x}_t|\mathbf{c})]dt + g(t)d\bar{\mathbf{w}}_t. \quad (3)$$

Under discrete time steps $k$, the update rule (e.g., Euler-Maruyama) explicitly showcases the refinement process:

$$\mathbf{x}_{k-1} \leftarrow \mathbf{x}_k + \eta\nabla_{\mathbf{x}} \log p_k(\mathbf{x}_k|\mathbf{c}) + \sigma\mathbf{z}. \quad (4)$$

The gradient term $\nabla \log p$ provides the **Manifold Adherence** force that continuously pulls the state back towards the manifold. However, standard implementations typically initialize from an uninformative prior $p_0 = \mathcal{N}(\mathbf{0}, \mathbf{I})$ ($t = 0$ as source and $t = 1$ as target). This implies the model must start from pure chaos and undergo extensive iterations to reach the manifold.

### 3.2. Residual Diffusion Bridge Formulation

To overcome the inefficiency of initializing from uninformative noise while retaining continuous precision, we adopt the **Diffusion Bridge** framework, which allows establishing probabilistic connections between any given source distribution $p_0$ and target data distribution $p_1$. We leverage Conditional Flow Matching (CFM) to instantiate this process,

aiming to learn the optimal transport path connecting $p_0$ and $p_1$. The path follows displacement interpolation:

$$\mathbf{x}_t = (1 - t)\mathbf{x}_0 + t\mathbf{x}_1 = \mathbf{x}_0 + t(\mathbf{x}_1 - \mathbf{x}_0). \quad (5)$$

This formulation provides the mathematical foundation for introducing our residual refinement paradigm, the velocity field $v_t$ that the model needs to learn is no longer a complex denoising field, but a constant residual vector:

$$v_t(\mathbf{x}_t) = \frac{d\mathbf{x}_t}{dt} = \mathbf{x}_1 - \mathbf{x}_0 \triangleq \Delta\mathbf{x}_{\text{residual}}. \quad (6)$$

The mathematical advantage of this perspective lies in the minimization of transport cost.

**Proposition 3.2** (Minimal Transport Cost). *Assuming the source distribution $p_0$ is anchored near the target manifold, i.e., $\mathbb{E}[\|\Delta\mathbf{x}_{residual}\|^2] \ll \mathbb{E}[\|\mathbf{x}_1\|^2]$, the kinetic transport cost required for residual bridging is significantly lower than that of generating from standard noise.*

This implies that the model only needs to learn a low-energy fine-tuning field, geometrically reducing learning difficulty.

### 3.3. Optimization Pathology: Loss Collapse

Although residual bridging guarantees low energy consumption geometrically, the optimization process may still fail if source distribution $p_0$ is chosen inappropriately. Specifically, setting $p_0$ as noise independent of condition $\mathbf{c}$ (i.e., $p_0 \perp \mathbf{c}$) exposes us to a critical issue known as **"Loss Collapse"**.

**Theorem 3.3** (Loss Collapse (Dong et al., 2025)). *If the source distribution $p_0(\mathbf{x})$ is independent of $\mathbf{c}$, the mutual information $I(\mathbf{x}_0; \mathbf{c}) = 0$. Consequently, as $t \to 0$, the true conditional vector field $u_t(\mathbf{x}|\mathbf{c})$ degenerates into the marginal vector field, causing the conditional gradient at the ground truth to vanish:*

$$\lim_{t \to 0} \mathbb{E}_{p_t(\mathbf{x}|\mathbf{c})}\left[\nabla_{\mathbf{c}}\|v_\theta(\mathbf{x}, t, \mathbf{c}) - u_t(\mathbf{x}|\mathbf{x}_1)\|^2\right] \approx 0. \quad (7)$$

This implies that initializing from noise can hinder the model's ability to attend to fine-grained instructions. It further indicates that a residual path alone is insufficient; the starting point itself should contain semantic information. To circumvent loss collapse, we construct a **Condition-Dependent Source** $p_0(\mathbf{x}|\mathbf{c})$. This serves not only to shorten the geometric distance but, more importantly, to inject non-trivial mutual information $I(\mathbf{x}_0; \mathbf{c}) > 0$ from the onset, thereby allowing gradient flow to capture semantic discrepancies. Consequently, establishing a semantic anchor is not merely a heuristic choice, but a theoretical prerequisite for preventing instruction drift in generative control policies.

## 4. ResVLA

Building upon the residual diffusion bridge perspective established in Sec.3, we introduce **ResVLA** (Figure 2). To

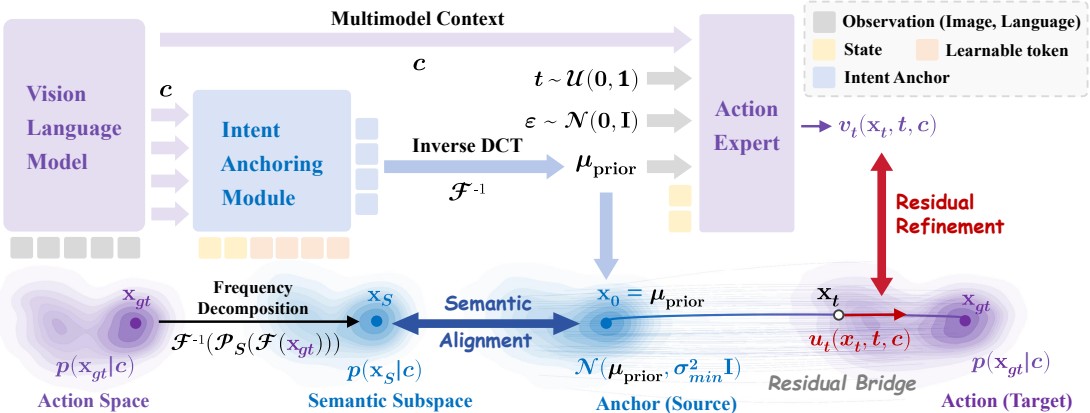

*Figure 2.* Overview of the **ResVLA** framework. The architecture consists of two cascading stages:(1) Intent Anchoring: The Intent Anchoring Module leverages VLM features to regress the low-frequency component $\mathbf{x}_{\mathcal{S}}$, constructing a condition-dependent source $p_0(\mathbf{x}|\mathbf{c})$. (2) Residual Bridging: A flow matching expert learns the residual transport path (red arrow) from this anchor to the full action $\mathbf{x}_{gt}$, focusing on refining high-frequency dynamics.

instantiate this framework, we seek physical counterparts for the mathematical "anchor" and "residual". ResVLA realizes this via spectral analysis: we disentangle action generation into a hierarchical bridging problem, first anchoring the low-frequency semantic intent, then refining high-frequency execution dynamics via flow matching.

**Frequency Decomposition:** To physically instantiate the decoupling of anchor and residual, we turn to **Frequency Analysis**. Physical intuition suggests that a robot's global intent (e.g., "approaching an object") manifests as long-horizon, smooth motion trends, whereas specific interaction details (e.g., "contact adjustment") appear as local, high-frequency jitter. This natural separation of physical characteristics aligns precisely with the mathematical structure we sought in Sec. 3. Let $\mathcal{F}$ denote the Discrete Cosine Transform (DCT). We partition the action space $\mathcal{X}$ into two complementary subspaces:

- **Semantic Subspace ($\mathcal{S}$):** Spanned by the lowest $k$ frequency modes, where $k$ is a learnable cutoff. It captures the **low-frequency global trajectory structure**, corresponding to the deterministic anchor.

- **Execution Subspace ($\mathcal{E}$):** As the orthogonal complement of $\mathcal{S}$, it captures **high-frequency detailed jitter**, corresponding to the stochastic residual.

Consequently, for any ground-truth action $\mathbf{x}_{gt}$, there exists a unique decomposition $\mathbf{x}_{gt} = \mathbf{x}_{\mathcal{S}} + \mathbf{x}_{\mathcal{E}}$. We model the semantic component as $\mathbf{x}_{\mathcal{S}} = \mathcal{F}^{-1}(\mathcal{P}_{\mathcal{S}}(\mathcal{F}(\mathbf{x}_{gt})))$, where $\mathcal{P}_{\mathcal{S}}$ is the low-pass projection operator that retains only the first $k$ spectral coefficients (Figure 2 bottom left).

**Intent Anchoring and Source Construction:** According to Theorem 1 (Loss Collapse), an effective refinement process must originate from a condition-dependent source. Given that $\mathbf{x}_{\mathcal{S}}$ carries deterministic semantic intent, our goal is to

learn a mapping from condition $\mathbf{c}$ to the semantic subspace $\mathcal{S}$ as the starting point for bridging. To this end, we propose the **Intent Anchoring Module** (Figure 2 middle). This module leverages the VLM backbone to extract semantic features and directly regresses the low-frequency component $\boldsymbol{\mu}_{\text{prior}}(\mathbf{c}) \approx \mathbf{x}_{\mathcal{S}}$ via a regression head. Centered on this prediction, we construct the source distribution of the diffusion bridge as:

$$p_0(\mathbf{x}|\mathbf{c}) = \mathcal{N}(\mathbf{x}; \boldsymbol{\mu}_{\text{prior}}(\mathbf{c}), \sigma_{\min}^2 \mathbf{I}). \qquad (8)$$

This distribution maximizes the mutual information $I(\mathbf{x}_0; \mathbf{c})$ at $t = 0$, providing a strongly semantically guided initialization for the subsequent refinement process.

**Residual Flow Matching:** Having established the semantic anchor, the refinement task transforms into injecting high-frequency details via the diffusion bridge. We employ residual flow matching to learn the transport path from prior $p_0$ to posterior $p_1$ (Figure 2 right). Following the residual dynamics in Sec. 3, the target vector field $u_t$ is dominated by the high-frequency component. Considering that $\mathbf{x}_0$ is sampled from the distribution centered at $\boldsymbol{\mu}_{\text{prior}}$, we have:

$$u_t(\mathbf{x}_t|\mathbf{x}_0, \mathbf{x}_{gt}) = \mathbf{x}_{gt} - \mathbf{x}_0 \approx \underbrace{\mathbf{x}_{\mathcal{E}}}_{\text{Refinement}} - \underbrace{\boldsymbol{\epsilon}}_{\text{Noise}}. \qquad (9)$$

This implies that the core task of the flow matching network $v_\theta$ is to fit this high-frequency residual $\mathbf{x}_{\mathcal{E}}$.

**Unified Optimization and Inference:** Our framework adopts an end-to-end bi-level optimization objective, strictly aligned with the spectral hierarchy:

$$\mathcal{L}_{\text{total}} = \lambda_{\text{sem}} \underbrace{\|\boldsymbol{\mu}_{\text{prior}} - \mathbf{x}_{\mathcal{S}}\|^2}_{\text{Semantic Alignment}} + \underbrace{\mathbb{E}_{t,\mathbf{x}_t} \|v_\theta - (\mathbf{x}_{gt} - \mathbf{x}_0)\|^2}_{\text{Residual Refinement}}$$
$$(10)$$

During inference, action generation follows a "Predict-Refine" flow. The model first predicts the semantic anchor

$\boldsymbol{\mu}_{\text{prior}}(\mathbf{c})$, and then evolves along the residual field via a numerical integrator:

$$\hat{\mathbf{x}} = \underbrace{\boldsymbol{\mu}_{\text{prior}}(\mathbf{c})}_{\text{Intent Anchor}} + \underbrace{\int_0^1 v_\theta(\mathbf{x}_t, t, \mathbf{c})\mathrm{d}t}_{\text{Iterative Refinement}}. \qquad (11)$$

Because the semantic anchor is already close to the target action, the residual transport path is short. Consequently, ResVLA can complete inference with significantly fewer function evaluations (NFE) than standard diffusion policies, substantially improving sampling efficiency.

## 5. Experiments

We design our experiments to empirically verify the three core hypotheses driving the **ResVLA** framework:

**H1: Semantic Anchoring and Robustness.** Does anchoring the generation process with a deterministic intent prior effectively mitigate semantic drift induced by loss collapse and enhance robustness against visual, linguistic, and layout perturbations in unstructured environments?

**H2: Learning Efficiency and Flow Straightness.** Does modeling sparse residual corrections, rather than global vector fields from noise, result in straighter probability flows that yield significantly faster training convergence and higher inference efficiency?

**H3: Cross-Embodiment and Real-World Validation.** Does separating low-frequency intent from high-frequency execution allow the model to distill embodiment-agnostic manipulation logic, thereby facilitating effective cross-embodiment transfer, and how does it perform in real-world manipulation?

### 5.1. Experimental Setup

**Benchmarks.** We evaluate ResVLA on four complementary settings: **LIBERO** (Liu et al., 2023a), its robustness-focused extension **LIBERO-Plus** (Fei et al., 2025), the cross-embodiment and realistic real-to-sim benchmark **SimplerEnv** (Li et al., 2024b), and a real-world **ALOHA** bimanual manipulation setup. Together, these environments rigorously assess the policy's capabilities across long-horizon sequencing, spatial reasoning, contact-rich manipulation, and out-of-distribution (OOD) generalization. Detailed specifications for each task are provided in Appendix B. **Crucially, our model evaluated across these benchmarks is trained entirely from scratch**.

**Baselines.** We compare **ResVLA** against a comprehensive set of state-of-the-art baselines categorized by their underlying control paradigms. First, we evaluate continuous generative policies, including foundation models such as Diffusion Policy, Octo, SmolVLA and GraspVLA, alongside recent high-performing architectures like GR00T N1, $\pi_0$ and the continuous variant of OpenVLA-OFT. Second, we benchmark against discrete autoregressive policies, comprising OpenVLA, discrete OpenVLA-OFT, SpatialVLA, ThinkAct, TraceVLA, NORA, UniVLA, UnifiedVLA and $\pi_0$-FAST. Finally, we encompass specialized paradigms integrating RL, Chain-of-Thought, World Models, or other advanced mechanisms, including GRAPE, VLA-RL, CoT-VLA, WorldVLA, VLA-OS, MolmoAct, FlowVLA, 4D-VLA, RIPT-VLA and PD-VLA.

**Metrics.** We evaluate: (1) **Success Rate (SR)** for task completion reliability; (2) **Learning Efficiency**, measured by performance under identical training steps; and (3) **Inference Efficiency**, assessing the trade-off between performance and the Number of Function Evaluations **(NFE)**.

### 5.2. Semantic Anchoring and Robustness (H1)

Our first hypothesis posits that the semantic prior mitigates semantic drift induced by loss collapse and improves robustness against environmental perturbations.

**Generalization vs. Semantic Over-fitting.** As summarized in Table 2, **ResVLA** achieves performance comparable to the state-of-the-art on the standard LIBERO benchmark. However, we argue that success on LIBERO alone does not fully capture a model's true capability, as this dataset is prone to semantic over-fitting, where policies memorize specific visual-textural correlations or fixed instruction phrasings. To assess adaptability, we evaluate ResVLA on the LIBERO-Plus benchmark (Table 1). While baselines like $\pi_0$ and OpenVLA-OFT exhibit sharp collapses under OOD visual and layout noise, ResVLA demonstrates exceptional resilience. Crucially, this robustness extends to linguistic variations: ResVLA maintains a dominant 88.5% success rate (↑7.5% over the best baseline) on diverse language instructions, whereas OpenVLA plummets to 23.0%. This confirms that anchoring generation with a deterministic intent prior prevents the policy from being misled by noise or synonymous phrasing, effectively mitigating the semantic drift observed in pure noise-to-action models.

**Stability in Unstructured Environments.** We further evaluate the model's robustness against physical and kinematic variations using the Robot and Layout settings in LIBERO-Plus. These settings simulate highly unstructured environments, introducing significant drift in agent morphology (embodiment) and spatial configuration relative to the training distribution. While diffusion-based baselines often exhibit instability in these out-of-distribution (OOD) scenarios, exemplified by $\pi_0$'s performance collapse to a mere 6.0% success rate in the Robot setting, ResVLA maintains a robust success rate of 59.9%. Simultaneously, in the Layout setting, ResVLA achieves a state-of-the-art success rate

*Table 1.* Robustness evaluation on the **LIBERO-Plus** benchmark. We report success rates (%) under various perturbations. The **best**, second-best, and third-best results are highlighted. For **ResVLA**, we report the performance gain (↑) or loss (↓) compared to the **best performing baseline**. See Appendix for detailed results per suite.

| Method | Original | Camera | Robot | Language | Light | Background | Noise | Layout | Total |
|---|---|---|---|---|---|---|---|---|---|
| *Reference: Trained on LIBERO-Plus (In-domain Adaptation)* | | | | | | | | | |
| **OpenVLA-OFT**(Kim et al., 2025) | 97.1 | 92.8 | 30.3 | 85.8 | 94.9 | 93.9 | 89.3 | 77.6 | 79.6 |
| *Trained on LIBERO (Generalization)* | | | | | | | | | |
| OpenVLA(Kim et al., 2024) | 76.5 | 0.8 | 3.5 | 23.0 | 8.1 | 34.8 | 15.2 | 28.5 | 15.6 |
| OpenVLA-OFT(Kim et al., 2025) | 97.1 | 56.4 | 31.9 | 79.5 | 88.7 | 93.3 | 75.8 | 74.2 | 69.6 |
| OpenVLA-OFT_w(Kim et al., 2025) | 95.3 | 10.4 | 38.7 | 70.5 | 76.8 | 93.6 | 49.9 | 69.9 | 55.8 |
| NORA(Hung et al., 2025) | 87.9 | 2.2 | 37.0 | 65.1 | 45.7 | 58.6 | 12.8 | 62.1 | 39.0 |
| WorldVLA(Cen et al., 2025) | 79.1 | 0.1 | 27.9 | 41.6 | 43.7 | 17.1 | 10.9 | 38.0 | 25.0 |
| UniVLA(Bu et al., 2025) | 95.2 | 1.8 | 46.2 | 69.6 | 69.0 | 81.0 | 21.2 | 31.9 | 42.9 |
| $\pi_0$(Black et al., 2026) | 94.2 | 13.8 | 6.0 | 58.8 | 85.0 | 81.4 | 79.0 | 68.9 | 53.6 |
| $\pi_0$-Fast(Pertsch et al., 2025) | 85.5 | **65.1** | 21.6 | 61.0 | 73.2 | 73.2 | 74.4 | 68.8 | 61.6 |
| RIPT-VLA(Tan et al., 2025) | 97.5 | 55.2 | 31.2 | 77.6 | 88.4 | 91.6 | 73.5 | 74.2 | 68.4 |
| OpenVLA-OFT_m(Kim et al., 2025) | 97.6 | 55.6 | 21.7 | 81.0 | 92.7 | 91.0 | 78.6 | 68.7 | 67.9 |
| **ResVLA (Ours)** | 96.6 | 53.2 ↓11.9 | **57.5** ↑11.3 | **88.2** ↑7.2 | **94.5** ↑1.8 | **96.3** ↑2.7 | **81.8** ↑2.8 | **78.3** ↑4.1 | **76.9** ↑7.3 |

*Table 2.* Comparison on LIBERO Benchmark. The **best**, second-best, and third-best results are highlighted. Our model is co-trained on all four task suites from **scratch** for 30k training steps without any pre-training.

| Method | Spatial | Object | Goal | Long | Avg. |
|---|---|---|---|---|---|
| Diffusion Policy† (Chi et al., 2025) | 78.3 | 92.5 | 68.3 | 50.5 | 72.4 |
| TraceVLA (Zheng et al., 2024) | 84.6 | 85.2 | 75.1 | 54.1 | 74.8 |
| Octo (Team et al., 2024) | 78.9 | 85.7 | 84.6 | 51.1 | 75.1 |
| OpenVLA (Kim et al., 2024) | 84.7 | 88.4 | 79.2 | 53.7 | 76.5 |
| SpatialVLA (Qu et al., 2025) | 88.2 | 89.9 | 78.6 | 55.5 | 78.1 |
| GRAPE (Zhang et al., 2024) | 87.6 | 91.2 | 82.2 | 55.8 | 79.2 |
| VLA-RL (Lu et al., 2025a) | 90.2 | 91.8 | 82.2 | 59.8 | 81.0 |
| CoT-VLA (Zhao et al., 2025) | 87.5 | 91.6 | 87.6 | 69.0 | 81.1 |
| WorldVLA (Cen et al., 2025) | 87.6 | 96.2 | 83.4 | 60.0 | 81.8 |
| ThinkAct (Huang et al., 2025) | 88.3 | 91.4 | 87.1 | 70.9 | 84.4 |
| $\pi_0$-FAST (Pertsch et al., 2025) | 96.4 | 96.8 | 88.6 | 60.2 | 85.5 |
| VLA-OS (Gao et al., 2025a) | 87.0 | 96.5 | 92.7 | 66.0 | 85.6 |
| MolmoAct (Lee et al., 2025) | 87.0 | 95.4 | 87.6 | 77.2 | 86.6 |
| NORA (Hung et al., 2025) | 92.2 | 95.4 | 89.4 | 74.6 | 87.9 |
| FlowVLA (Zhong et al., 2025b) | 93.2 | 95.0 | 91.6 | 72.6 | 88.1 |
| 4D-VLA (Zhang et al., 2025) | 88.9 | 95.2 | 90.9 | 79.1 | 88.6 |
| SmolVLA (Shukor et al., 2025) | 93.0 | 94.0 | 91.0 | 77.0 | 88.8 |
| GraspVLA (Deng et al., 2025) | - | 94.1 | 91.2 | 82.0 | 89.1 |
| GR00T N1 (Bjorck et al., 2025) | 94.4 | 97.6 | 93.0 | 90.6 | 93.9 |
| $\pi_0$ (Black et al., 2026) | 96.8 | 98.8 | 95.8 | 85.2 | 94.2 |
| PD-VLA (Song et al., 2025) | 95.5 | 96.7 | 94.9 | 91.7 | 94.7 |
| UniVLA (Bu et al., 2025) | 96.5 | 96.8 | 95.6 | 92.0 | 95.2 |
| UnifiedVLA (Wang et al., 2025c) | 95.4 | 98.8 | 93.6 | 94.0 | 95.5 |
| OpenVLA-OFT (Kim et al., 2025) | 97.6 | 98.4 | **97.9** | 94.5 | **97.1** |
| VLA-Adapter (Wang et al., 2025b) | **97.8** | **99.2** | 97.2 | **95.0** | **97.3** |
| **ResVLA (Ours)** | 96.0 | **100.0** | 97.4 | 92.8 | 96.6 |

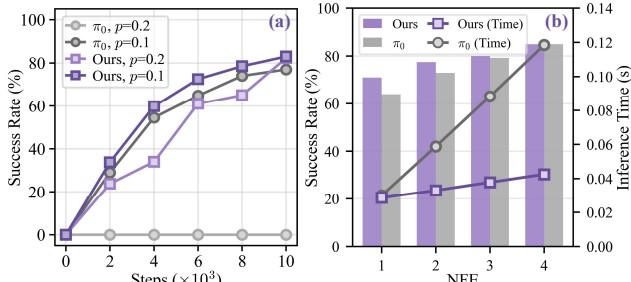

*Figure 3.* **Efficiency Evaluation.** (a) Training convergence curves comparing success rates under varying dropout rates ($p$). (b) Inference analysis displaying success rates (bars) and inference time (lines) across different numbers of function evaluations (NFE).

### 5.3. Learning Efficiency and Flow Straightness (H2)

We investigate whether residual corrections yield straighter flows for improved efficiency, benchmarking against $\pi_0$ (starVLA Contributors, 2025) on LIBERO.

**Training Convergence.** Figure 3(a) demonstrates ResVLA's superior sample efficiency. Under standard dropout rate ($p = 0.1$), it converges significantly faster than the baseline. Crucially, in high dropout rate ($p = 0.2$), ResVLA maintains robust performance ascent while $\pi_0$ suffers from optimization saturation. This confirms that spectral decomposition anchors the learning process, effectively alleviating the optimization burden inherent in noise-to-action mapping.

**Inference Efficiency.** Figure 3(b) highlights the structural advantage of our approach. While both models reach ~85% success at NFE=4, ResVLA achieves > 70% with a single step (NFE=1). This validates our *Path Straightening* hypothesis: initiating transport from a task-aligned anchor minimizes curvature, yielding linear trajectories for rapid manifold traversal. Consequently, the Residual Diffusion

of 79.0%. This confirms that frequency-domain modeling provides reliable global guidance for spatial reasoning, effectively suppressing the trajectory jitter and execution errors common in generative policies under OOD conditions, while ensuring precise and contact-rich manipulation. Camera perturbations remain comparatively challenging, indicating that the model still inherits viewpoint sensitivity from the training data, while training on a single dataset alone is insufficient to fully address generalization under camera viewpoint changes.

*Table 3.* **Performance Comparison on SimplerEnv (Google Robot).** Four tasks: Pick Coke Can (**PC**), Move Near (**MN**), Open Drawer (**OD**), and Open Top Drawer (**OTD**). The **best**, second-best, and third-best results are highlighted (excluding baselines with spatial co-training). Results show that our **ResVLA**, learned entirely from scratch, achieves competitive performance.

| Models | Pre-Train | PC | MN | OD | OTD | Avg. |
|---|---|---|---|---|---|---|
| *Baselines with Spatial Co-training* | | | | | | |
| RT-2-X (Brohan et al., 2023a) | ✓ | 78.7 | 77.9 | 25.0 | 3.7 | 46.3 |
| Magma (Yang et al., 2025) | ✓ | 83.7 | 65.4 | 56.0 | 6.4 | 52.9 |
| InternVLA-M1 (Contributors, 2025) | ✓ | 95.3 | 90.0 | 75.5 | 62.0 | 80.7 |
| *Baselines without Spatial Co-training* | | | | | | |
| RT-1 (Brohan et al., 2023b) | ✓ | 85.7 | 44.2 | **73.0** | 6.5 | 52.4 |
| RT-1-X (O'Neill et al., 2024) | ✓ | 56.7 | 31.7 | 59.7 | 21.3 | 42.4 |
| OpenVLA (Kim et al., 2024) | ✓ | 18.0 | 56.3 | 63.0 | 0.0 | 34.3 |
| CogACT (Li et al., 2024a) | ✓ | **91.3** | **85.0** | 71.8 | 50.9 | 74.8 |
| SpatialVLA (Qu et al., 2025) | ✓ | 86.0 | 77.9 | 57.4 | - | 75.1 |
| $\pi_0$ (Black et al., 2026) | ✓ | 72.7 | 65.3 | 38.3 | - | 58.8 |
| $\pi_0$-FAST (Pertsch et al., 2025) | ✓ | 75.3 | 67.5 | 42.9 | - | 61.9 |
| GR00T N1.5 (Bjorck et al., 2025) | ✓ | 51.7 | 54.0 | 27.8 | 7.4 | 35.2 |
| InternVLA-M1 (Vanilla) | ✓ | 90.0 | 69.8 | 52.5 | **52.2** | 66.1 |
| **ResVLA (Ours)** | ✗ | 87.0 | 78.8 | 45.8 | 37.0 | 62.2 |

*Table 4.* **Performance Comparison on SimplerEnv (WidowX/Bridge).** We evaluate models across four tasks: Spoon on Towel (**ST**), Carrot on Plate (**CP**), Stack Blocks (**SB**), and Eggplant in Basket (**EB**). The **best**, second-best, and third-best results are highlighted. Results show our **ResVLA** achieves competitive performance, particularly in the Eggplant task, despite lacking large-scale pre-training.

| Models | Pre-Train | ST | CP | SB | EB | Avg. |
|---|---|---|---|---|---|---|
| *Baselines with Spatial Co-training* | | | | | | |
| Magma (Yang et al., 2025) | ✓ | 37.5 | 31.0 | 12.7 | 60.5 | 35.8 |
| InternVLA-M1 (Contributors, 2025) | ✓ | 87.5 | 67.9 | 31.3 | 100.0 | 71.7 |
| *Baselines without Spatial Co-training* | | | | | | |
| RT-1-X (Brohan et al., 2023b) | ✓ | 0.0 | 4.2 | 0.0 | 0.0 | 1.1 |
| Octo-Base (Team et al., 2024) | ✓ | 15.8 | 12.5 | 0.0 | 41.7 | 17.5 |
| Octo-Small (Team et al., 2024) | ✓ | 41.7 | 8.2 | 0.0 | 56.7 | 26.7 |
| OpenVLA (Kim et al., 2024) | ✓ | 4.2 | 0.0 | 0.0 | 12.5 | 4.2 |
| CogACT (Li et al., 2024a) | ✓ | 71.7 | 50.8 | 15.0 | 67.5 | 51.3 |
| SpatialVLA (Qu et al., 2025) | ✓ | 16.7 | 25.0 | 29.2 | **100.0** | 42.7 |
| $\pi_0$ (Black et al., 2026) | ✓ | 29.1 | 0.0 | 16.6 | 62.5 | 27.1 |
| $\pi_0$-FAST (Pertsch et al., 2025) | ✓ | 29.1 | 21.9 | 10.8 | 66.6 | 48.3 |
| GR00T N1.5 (Bjorck et al., 2025) | ✓ | 75.3 | 54.3 | 57.0 | 61.3 | 61.9 |
| **ResVLA (Ours)** | ✗ | **89.6** | 50.0 | 38.5 | 96.9 | **68.8** |

*Table 5.* **Real-robot evaluation on ALOHA.** Stage-wise success rates over 10 trials.

| Method | Pick Cup (%) | Handover (%) | Placement / Overall (%) |
|---|---|---|---|
| $\pi_{0.5}$ | 50 | 40 | 10 |
| **ResVLA** | **60** | **50** | **20** |

Bridge ensures significantly lower latency scaling compared to standard iterative denoising.

### 5.4. Cross-Embodiment Generalization and Real-World Validation(H3)

Our third hypothesis examines whether ResVLA can distill embodiment-agnostic manipulation logic that facilitates cross-embodiment transfer, and whether the same refinement paradigm remains effective in real-world manipulation.

**Results on SimplerEnv.** We evaluate cross-embodiment transfer via **co-training** experiments on the SimplerEnv benchmark. As shown in Table 3 and Table 4, despite being trained **entirely from scratch** without large-scale robot pre-training or specialized spatial co-training optimizations, ResVLA demonstrates strong cross-embodiment generalization. On the Google Robot suite, ResVLA achieves an average success rate of **68.4%**, outperforming prominent pre-trained baselines including $\pi_0$ (58.8%), OpenVLA (34.3%), and RT-1-X (42.4%). On the WidowX suite, ResVLA achieves an average success rate of **57.9%**, outperforming RT-1-X (1.1%), Octo-Base (17.5%), OpenVLA (4.2%), CogACT (51.3%), and SpatialVLA (42.7%), while remaining competitive with stronger spatially co-trained baselines. These results suggest that ResVLA captures transferable manipulation structure while preserving embodiment-specific local execution refinement.

**Real-World Evaluation.** To further assess whether this refinement paradigm transfers beyond simulation, we conduct real-robot experiments on an ALOHA bimanual manipulation platform. We consider a contact-rich three-stage task consisting of *Pick Cup*, *Handover*, and *Placement*. In each episode, one arm first grasps a cup from the table, then transfers it stably to the other arm, and finally the receiving arm places the cup onto a cup stand. This task is challenging because errors accumulate across stages: grasp quality affects handover stability, and handover errors further propagate to final placement accuracy. As shown in Figure 4 and Table 5, ResVLA achieves competitive real-world performance on this task.

## 6. Conclusion

In this work, we challenged the "Generation-from-Noise" orthodoxy in VLA modeling. Building on recent analyses of source-condition independence and loss collapse, we proposed **ResVLA**, a framework grounded in the **Residual Diffusion Bridge** perspective. By decomposing generation into *semantic anchoring* and *residual refinement*, we reduced the complexity of the learning problem and shortened the transport path. Our results confirm that physics-aware structural bias, specifically the separation of intent and execution, yields substantial gains in performance, efficiency, and robustness.

## 7. Limitations and Future Work

Despite the efficiency and robustness demonstrated by **ResVLA**, limitations provide avenues for future research.

**The Diversity of Anchor Choices.** First, our choice of low-frequency components as intent anchors is motivated by their physical clarity and ease of supervision. However, we emphasize that this is merely one instantiation of

| Pick Cup | Handover | Placement |

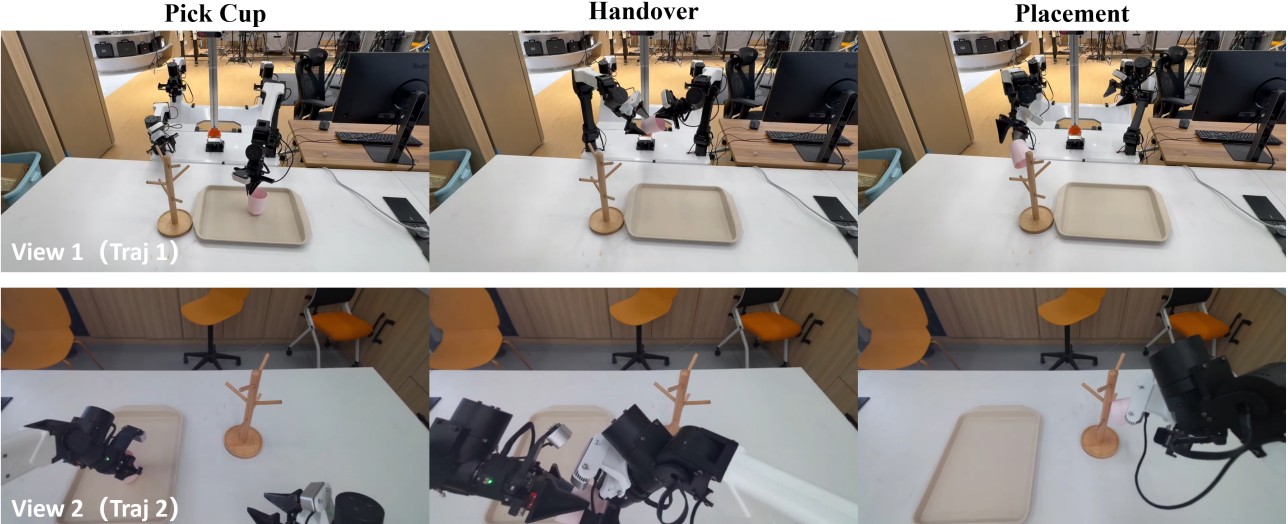

*Figure 4.* Visualization of successful executions from two camera viewpoints and two different episodes, illustrating the full task pipeline: *Pick Cup → Handover → Placement.* The task requires tight dual-arm coordination and is susceptible to stage-to-stage error accumulation.

the **"Residual Diffusion Bridge"** framework. For specific manipulation tasks, the frequency domain may not be the unique or optimal decoupling space. Future work could explore alternative structured anchors; for instance, it is feasible to leverage **coarse-grained discrete action tokens** derived from large models as the initialization point.

**Verification at Scale.** Second, constrained by computational resources, ResVLA has currently been validated on a subset of mainstream benchmarks and has not yet undergone full-scale pre-training on massive, open-domain datasets like Open X-Embodiment. Consequently, the **Scaling Law** of this architecture under significant expansion of model parameters and data scale remains to be empirically investigated. Nonetheless, the high sample efficiency demonstrated by ResVLA suggests its promise as an ideal paradigm for efficient fine-tuning of large foundation models. Future efforts will focus on integrating ResVLA into larger-scale pre-training pipelines to verify its scalability in generalist robotic control.

## Acknowledgements

This work was supported by the National Natural Science Foundation of China (Project Number 62595774), Shanghai Frontiers Science Center of Human-centered Artificial Intelligence (ShangHAI), MoE Key Laboratory of Intelligent Perception and Human-Machine Collaboration (KLIP-HuMaCo).

## Impact Statement

This work advances the precision and robustness of Vision-Language-Action (VLA) models, potentially accelerating the deployment of generalist robots in unstructured domestic and industrial environments. While our "Refinement-from-Intent" paradigm aims to reduce erratic behaviors and improve instruction adherence, the integration of large generative models into physical control systems introduces inherent safety risks, such as unintended physical interactions arising from VLM hallucinations or distribution shifts. We emphasize the importance of implementing rigorous safety guardrails and hardware-level constraints alongside such learning-based policies.

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

## A. Proofs and Derivations

In this section, we provide detailed derivations for the theoretical propositions and theorems presented in the main text.

### A.1. Proof of Proposition 3.1 (Irreducible Quantization Noise)

**Proposition 1.** *For a uniform quantization $\mathcal{Q}$ with resolution $\Delta$, assuming locally uniform ground-truth $\mathbf{x}^*$, the expected reconstruction MSE is strictly lower-bounded by $\frac{\Delta^2}{12}$.*

*Proof.* Consider a 1-dimensional continuous signal $x \in \mathbb{R}$. The uniform quantizer $\mathcal{Q}$ maps $x$ to the nearest discrete bin center $c_k$. The quantization error is defined as $e = x - \mathcal{Q}(x)$.

For a quantization grid with resolution $\Delta$, the error $e$ is bounded within the interval $[-\frac{\Delta}{2}, \frac{\Delta}{2}]$. Under the assumption that the ground-truth signal $x^*$ is locally uniformly distributed within the bin (a standard high-resolution assumption in signal processing), the error $e$ follows a uniform distribution:

$$p(e) = \begin{cases} \frac{1}{\Delta} & \text{if } -\frac{\Delta}{2} \leq e \leq \frac{\Delta}{2} \\ 0 & \text{otherwise} \end{cases} \tag{12}$$

The Mean Squared Error (MSE) corresponds to the variance of this uniform distribution. We calculate the expectation:

$$\mathbb{E}[e^2] = \int_{-\infty}^{\infty} e^2 p(e)\, de = \int_{-\frac{\Delta}{2}}^{\frac{\Delta}{2}} e^2 \cdot \frac{1}{\Delta}\, de. \tag{13}$$

Solving the integral:

$$\begin{aligned} \mathbb{E}[e^2] &= \frac{1}{\Delta} \left[ \frac{e^3}{3} \right]_{-\frac{\Delta}{2}}^{\frac{\Delta}{2}} \\ &= \frac{1}{3\Delta} \left( \frac{\Delta^3}{8} - \left( -\frac{\Delta^3}{8} \right) \right) \\ &= \frac{1}{3\Delta} \left( \frac{\Delta^3}{4} \right) = \frac{\Delta^2}{12}. \end{aligned} \tag{14}$$

For a high-dimensional action vector $\mathbf{x} \in \mathbb{R}^D$ where each dimension is quantized independently, the total expected squared error matches this variance per dimension (or scales linearly with $D$ if summing). Thus, the irreducible error floor per dimension is $\frac{\Delta^2}{12}$. $\square$

### A.2. Proof of Proposition 2 (Minimal Transport Cost)

**Proposition 2.** *Assuming the source distribution $p_0$ is anchored near the target manifold, i.e., $\mathbb{E}[\|\Delta\mathbf{x}_{residual}\|^2] \ll \mathbb{E}[\|\mathbf{x}_1\|^2]$, the kinetic transport cost required for residual bridging is significantly lower than that of generating from standard noise.*

*Proof.* In Optimal Transport (OT) and Flow Matching frameworks, the *Kinetic Transport Cost* (or energy) of a trajectory is defined as the integral of the squared velocity norm:

$$\mathcal{C} = \int_0^1 \|v_t(\mathbf{x}_t)\|^2\, dt. \tag{15}$$

Modern Flow Matching objectives encourage straight transport paths (constant velocity). For a linear path connecting a source $\mathbf{x}_0$ and a target $\mathbf{x}_1$, the velocity is constant: $v_t = \mathbf{x}_1 - \mathbf{x}_0$. The cost simplifies to the squared Euclidean distance:

$$\mathcal{C} = \|\mathbf{x}_1 - \mathbf{x}_0\|^2. \tag{16}$$

We compare the expected cost of standard methods versus our residual approach:

- **Case 1: Standard Diffusion/Flow.** The source $\mathbf{x}_0 \sim \mathcal{N}(\mathbf{0}, \mathbf{I})$ is isotropic Gaussian noise. The expected cost is proportional to the total signal energy plus the noise variance:

$$\mathbb{E}[\mathcal{C}_{\text{standard}}] = \mathbb{E}[\|\mathbf{x}_1 - \mathbf{x}_{\text{noise}}\|^2] \approx \mathbb{E}[\|\mathbf{x}_1\|^2] + \mathbb{E}[\|\mathbf{x}_{\text{noise}}\|^2]. \tag{17}$$

- **Case 2: Residual Bridging (Ours).** The source $\mathbf{x}_0 = \mathbf{x}_{\text{anchor}}$ is the semantic anchor predicted by the VLA. The flow models only the residual vector $\Delta\mathbf{x} = \mathbf{x}_1 - \mathbf{x}_{\text{anchor}}$. The expected cost is:

$$\mathbb{E}[\mathcal{C}_{\text{residual}}] = \mathbb{E}[\|\mathbf{x}_1 - \mathbf{x}_{\text{anchor}}\|^2] = \mathbb{E}[\|\Delta\mathbf{x}\|^2]. \tag{18}$$

**Conclusion.** Since the VLA is trained to capture the primary mode of the target distribution, the anchor $\mathbf{x}_{\text{anchor}}$ lies in the local neighborhood of the ground truth $\mathbf{x}_1$. This ensures that the residual magnitude is significantly smaller than the absolute signal magnitude: $\mathbb{E}[\|\Delta\mathbf{x}\|^2] \ll \mathbb{E}[\|\mathbf{x}_1\|^2]$. Consequently, we derive the inequality:

$$\mathbb{E}[\mathcal{C}_{\text{residual}}] \ll \mathbb{E}[\mathcal{C}_{\text{standard}}]. \tag{19}$$

This implies the residual vector field has a significantly lower magnitude, resulting in simpler dynamics that are easier to learn and numerically integrate. $\square$

### A.3. Derivation of Theorem 3 (Loss Collapse)

**Theorem 3.** *As $t \to 0$, if the initial distribution $p_0(\mathbf{x})$ is independent of $\mathbf{c}$, the conditional vector field degenerates, causing the supervision gradients to vanish.*

*Proof.* We analyze the behavior of the vector field through the lens of the probability path it generates. Consider the conditional vector field $u_t(\mathbf{x}|\mathbf{c})$ which drives the flow of the probability density $p_t(\mathbf{x}|\mathbf{c})$. In diffusion and flow matching models, the vector field is intrinsically linked to the score function of the density: $u_t(\mathbf{x}|\mathbf{c}) \propto \nabla_{\mathbf{x}} \log p_t(\mathbf{x}|\mathbf{c})$.

At $t = 0$, standard initialization samples $\mathbf{x}_0$ from a pure Gaussian prior $\mathcal{N}(\mathbf{0}, \mathbf{I})$, which is statistically independent of the condition $\mathbf{c}$. Mathematically, the conditional density degenerates to the prior:

$$p_0(\mathbf{x}|\mathbf{c}) = p_{\text{prior}}(\mathbf{x}) = \mathcal{N}(\mathbf{x}; \mathbf{0}, \mathbf{I}). \tag{20}$$

Since the density at $t = 0$ is identical for all conditions $\mathbf{c}$, the geometric structure of the vector field generating this density must also be independent of $\mathbf{c}$. Specifically, the score function becomes unconditional:

$$\nabla_{\mathbf{x}} \log p_0(\mathbf{x}|\mathbf{c}) = \nabla_{\mathbf{x}} \log p_{\text{prior}}(\mathbf{x}). \tag{21}$$

Consequently, the optimal vector field $u_0^*(\mathbf{x}|\mathbf{c})$ at the initial boundary contains no information about the target task. The gradient of the learning objective $\mathcal{L}_{\text{CFM}}$ with respect to the condition $\mathbf{c}$ vanishes:

$$\mathbb{E}\left[\nabla_{\mathbf{c}} \|v_\theta(\mathbf{x}_0, 0, \mathbf{c}) - u_0^*(\mathbf{x}|\mathbf{c})\|^2\right] \approx 0. \tag{22}$$

This phenomenon, termed "Loss Collapse," implies that in the initial phase of generation, the model receives no effective gradient signal to distinguish between different tasks $\mathbf{c}$, as the supervision signal is dominated by the task-agnostic noise structure. $\square$

## B. Benchmark Specifications and Detailed Results

This section details the experimental setups, task definitions, and comprehensive per-task results referenced in Section 5. Our goal is to provide full transparency into the evaluation protocols that validate the efficacy of our 'Refinement-from-Intent' framework.

### B.1. LIBERO and LIBERO-Plus Suite

**(1) Standard Task Suites.** We utilize the official LIBERO benchmark (Liu et al., 2023a) as our primary evaluation domain for long-horizon and knowledge transfer capabilities. The benchmark is stratified into **four** specific suites:

- **LIBERO-Spatial (10 tasks):** Tests the agent's ability to generalize spatial relationships (e.g., "pick up the bowl next to the plate"). The object instances remain constant, but their relative layouts vary.

- **LIBERO-Object (10 tasks):** Evaluates robustness to visual object variations. The layout is fixed, but the target objects change (e.g., picking up a red mug vs. a white mug).

- **LIBERO-Goal (10 tasks):** Focuses on procedural generalization where the scene is fixed, but the goal instruction directs the robot to perform different manipulations (e.g., "open the top drawer" vs. "open the bottom drawer").

- **LIBERO-Long (10 tasks):** The most challenging suite, requiring the execution of long-horizon sequences (e.g., retrieve an object, navigate, and place it), often involving 50+ simulation steps.

A visual overview of representative tasks from these four evaluation domains is provided in Figure 5.

### LIBERO-Spatial

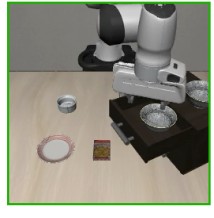 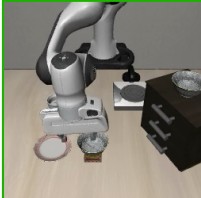

**Spatial Task Instruction**
1. pick up the black bowl next to the cookie box and place it on the plate
2. pick up the black bowl in the top drawer of the wooden cabinet and place it on the plate
3. pick up the black bowl on the ramekin and place it on the plate
4. pick up the black bowl on the stove and place it on the plate
5. pick up the black bowl between the plate and the ramekin and place it on the plate
6. pick up the black bowl on the cookie box and place it on the plate
7. pick up the black bowl next to the plate and place it on the plate
8. pick up the black bowl next to the ramekin and place it on the plate
9. pick up the black bowl from table center and place it on the plate
10. pick up the black bowl on the wooden cabinet and place it on the plate

### LIBERO-Object

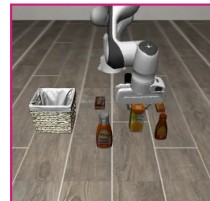 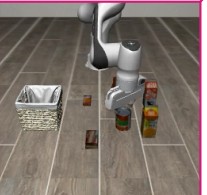

**Object Task Instruction**
1. pick up the orange juice and place it in the basket
2. pick up the ketchup and place it in the basket
3. pick up the cream cheese and place it in the basket
4. pick up the bbq sauce and place it in the basket
5. pick up the alphabet soup and place it in the basket
6. pick up the milk and place it in the basket
7. pick up the salad dressing and place it in the basket
8. pick up the butter and place it in the basket
9. pick up the tomato sauce and place it in the basket
10. pick up the chocolate pudding and place it in the basket

### LIBERO-Goal

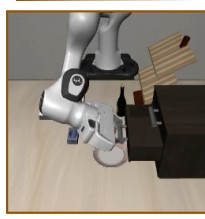 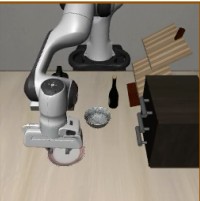

**Goal Task Instruction**
1. put the bowl on the plate
2. put the bowl on the stove
3. put the bowl on top of the cabinet
4. put the wine bottle on the rack
5. put the wine bottle on top of the cabinet
6. turn on the stove
7. put the cream cheese in the bowl
8. push the plate to the front of the stove
9. open the middle drawer of the cabinet
10. open the top drawer and put the bowl inside

### LIBERO-Long

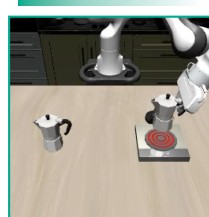 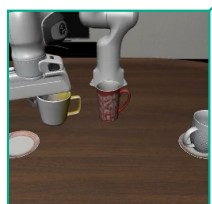

**Long Task Instruction**
1. put the white mug on the left plate and put the yellow and white mug on the right plate
2. put the white mug on the plate and put the chocolate pudding to the right of the plate
3. put the yellow and white mug in the microwave and close it
4. turn on the stove and put the moka pot on it
5. put both the alphabet soup and the cream cheese box in the basket
6. put both the alphabet soup and the tomato sauce in the basket
7. put both moka pots on the stove
8. put both the cream cheese box and the butter in the basket
9. put the black bowl in the bottom drawer of the cabinet and close it
10. pick up the book and place it in the back compartment of the caddy

*Figure 5.* **Visualization of the standard LIBERO benchmark suites.** We illustrate representative task initializations and instructions from the four evaluation domains: **LIBERO-Spatial** (spatial layout generalization), **LIBERO-Object** (visual object generalization), **LIBERO-Goal** (procedural goal generalization), and **LIBERO-Long** (long-horizon sequential planning).

*Table 6.* Detailed robustness breakdown across different LIBERO-Plus task suites. We report success rates (%) on seven distinct perturbation axes and the overall average.

| Task | Camera | Robot | Language | Light | Background | Noise | Layout | Total |
|---|---|---|---|---|---|---|---|---|
| **LIBERO-Spatial** | 58.5 | 71.4 | 98.7 | 97.9 | 98.1 | 88.0 | 93.8 | 85.9 |
| **LIBERO-Object** | 62.6 | 43.0 | 88.1 | 99.7 | 99.6 | 93.6 | 86.4 | 80.1 |
| **LIBERO-Goal** | 60.0 | 55.7 | 74.6 | 99.6 | 98.2 | 86.8 | 60.0 | 74.0 |
| **LIBERO-Long** | 32.7 | 61.6 | 91.9 | 79.9 | 90.0 | 61.7 | 73.7 | 68.2 |
| **Average** | 53.2 | 57.5 | 88.2 | 94.5 | 96.3 | 81.8 | 78.3 | 76.9 |

**(2) LIBERO-Plus Perturbation Protocols.** To rigorously assess robustness beyond standard generalization, we evaluate ResVLA on the **LIBERO-Plus** benchmark (Fei et al., 2025). Following the definitions in the official documentation, we evaluate performance across **seven** distinct perturbation dimensions, each containing specific sub-dimensions:

- **Objects Layout:** Designed to test robustness against object-level disturbances. This includes: (1) *Confounding Objects*, where unseen distractor objects are randomly added to the scene; and (2) *Target Object Pose*, where the target object's initial position and orientation are randomized while maintaining essential semantic relations.

- **Background Textures:** Evaluates generalization to environmental appearance changes. Sub-dimensions include: (1) *Scene Theme*, which modifies the texture of the environment walls (e.g., brick, painted); and (2) *Surface Appearance*, which alters the texture of the working surface (e.g., tabletop or floor).

- **Light Conditions:** Tests visual understanding under varying illumination defined by four parameters: *Diffuse* color (reflected light), *Direction* of the light source, *Specular* intensity (highlights on surfaces), and the presence of cast *Shadows*.

- **Camera Viewpoints:** Tests view-invariant representation by modifying camera parameters: (1) *Camera Distance* (scaling along the optical axis); (2) *Spherical Position* (altering azimuth and elevation on a sphere centered at the scene); and (3) *Camera Orientation* (perturbing yaw, pitch, and roll).

- **Robot Initial States:** Applies random perturbations to the robot arm's *Initial Joint Angles* ($q_{pos}$), with perturbation magnitudes ranging from 0.1 to 0.5, testing the policy's ability to recover from varied starting configurations.

- **Language Instructions:** Utilizes LLMs to rewrite instructions into three variants: (1) *Distraction* (adding conversational, task-irrelevant context); (2) *Common Sense* (replacing object names with functional descriptions); and (3) *Reasoning Chain* (altering reasoning complexity).

- **Sensor Noise:** Simulates real-world sensor imperfections to evaluate robustness under degraded input quality. This includes five noise types: *Motion Blur*, *Gaussian Blur* (defocus), *Zoom Blur*, *Fog* (atmospheric interference), and *Glass Blur* (distortion and refraction).

We visually illustrate the severity of these seven perturbation dimensions in Figure 6.

To quantify our model's resilience against these severe distribution shifts, we present a fine-grained performance breakdown. Table 6 details the success rates across all four LIBERO task suites for each of the seven perturbation axes defined above.

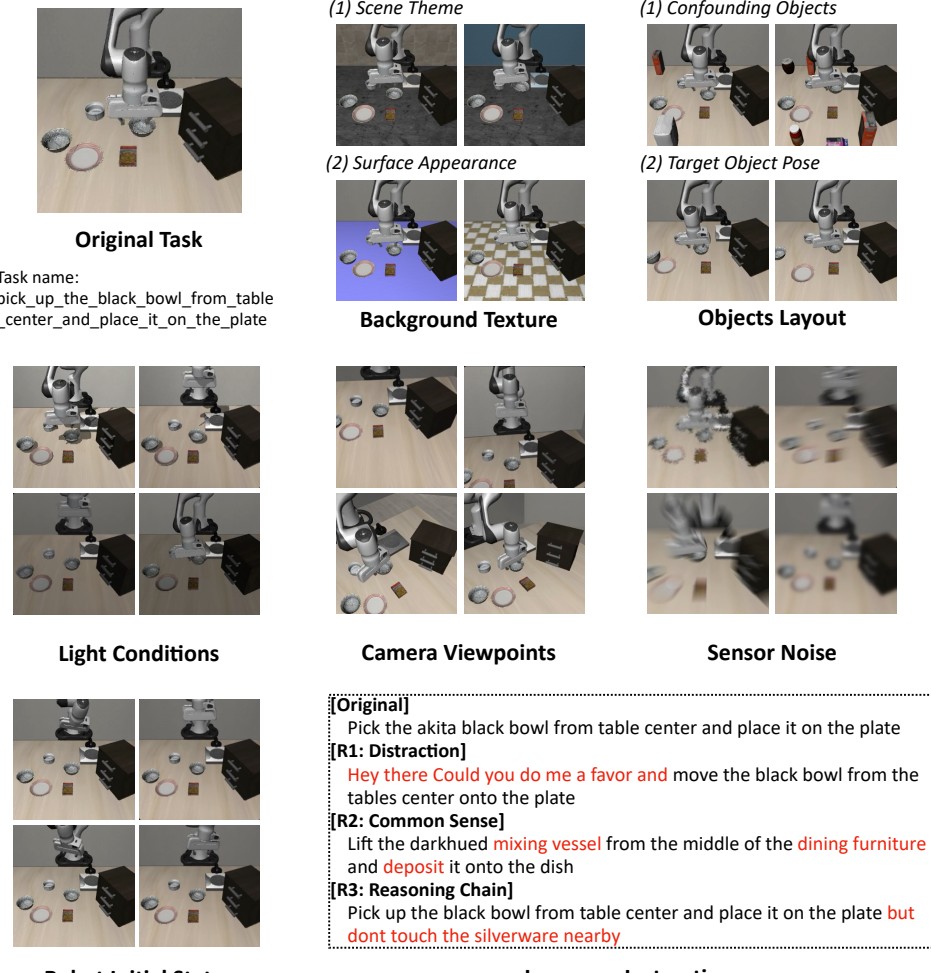

*Figure 6.* **Visualization of the seven perturbation dimensions in the LIBERO-Plus benchmark.** We showcase the task "*pick up the black bowl from table center and place it on the plate*" from the LIBERO-Spatial suite. The central image represents the **Original Task** (standard LIBERO). Surrounding panels illustrate the significant distribution shifts introduced by LIBERO-Plus, including visual variations (e.g., **Background Texture** split into scene themes/surfaces, **Sensor Noise**, **Light Conditions**), geometric changes (**Objects Layout**, **Camera Viewpoints**, **Robot Initial States**), and semantic shifts. The **Language Instructions** panel highlights adversarial rewrites, with red text denoting distractors, synonym replacements, and reasoning constraints. These visualizations highlight the severity of the domain gaps that our model successfully overcomes (quantitative breakdown provided in Table 6).

## B.2. SimplerEnv Task Descriptions

For the cross-embodiment experiments reported in Table 3 and 4, we utilize **SIMPLER** (Li et al., 2024b), a simulated evaluation framework designed to proxy real-world performance for Google Robot and WidowX embodiments.

**(1) Google Robot (Table 3):** We utilize the Fractal20220817 dataset. We evaluate on four tasks involving pick-and-place, spatial rearrangement, and articulated object manipulation:

- **Pick Coke Can (*Pick Coke*):** The robot must grasp and lift a Coke can. The can is initialized in three distinct orientations (standing, lying horizontally, lying vertically) across 25 different grid positions on the table (75 trials in total).

- **Move obj1 Near obj2 (*Move Near*):** This task requires moving a specified source object near a target object in the presence of a distractor. The evaluation rotates through 5 different object triplets (selected from 8 common items: blue plastic bottle, pepsi can, orange, 7up can, apple, sponge, coke can, redbull can) arranged in triangle patterns to test

semantic understanding and obstacle avoidance (60 trials in total).

- **(Open / Close) (Top / Middle / Bottom) Drawer (*Open Drawer*):** The robot is tasked with opening or closing a specific drawer (top, middle or bottom) of a cabinet. This assesses the ability to manipulate articulated objects. The robot's base position is varied across 9 locations (54 trials in total).

- **Open Top Drawer and Place Apple (*Open Top Drawer*):** A long-horizon sequential task where the robot must first open the top drawer and then pick up an apple from the cabinet surface to place it inside. The language instruction updates from "open" to "place" during the episode (27 trials in total).

**(2) WidowX + Bridge (Table 4)**: We utilize the BridgeData V2 dataset. The benchmark consists of four manipulation tasks, each evaluated over 24 trials:

- **Put the Spoon on the Towel (*Spoon on Towel*):** The robot must grasp a spoon and place it on a towel. The objects are initialized at the vertex of a 15cm square. The spoon's initial orientation varies between horizontal and vertical, requiring the agent to perform precise gripper re-orientation (24 trials in total).

- **Put Carrot on Plate (*Carrot on Plate*):** This task shares the same 15cm geometric layout as the above spoon task but substitutes the objects with a carrot and a plate (24 trials in total).

- **Stack the Green Block on the Yellow Block (*Stack Blocks*):** The goal is to stack a 3cm green cube onto a yellow cube(these two cubes are placed at the square vertex). The task includes two different modes based on the initialization distance: a closer setting (10cm) and a farther setting (20cm), with 12 trials for each (24 trials in total).

- **Put Eggplant into Yellow Basket (*Eggplant in Basket*):** This task requires moving an eggplant from the right basin in a sink to a yellow basket in the left basin. The eggplant is initialized with random positions and orientations to test robustness (24 trials in total).

## C. Implementation Details and Hyperparameters

### C.1. Implementation Details

**Architecture.** We instantiate ResVLA using the Qwen-VL architecture as the vision-language backbone. Specifically, we utilize the 2B parameter variant (Qwen3-2B) to balance semantic understanding with inference latency. The VLM backbone is fine-tuned using LoRA or full fine-tuning depending on the dataset scale, while the frequency-aware action head is trained from scratch.

**Hardware and Infrastructure.** To ensure consistent evaluation, all models were trained on a single compute node equipped with **4 NVIDIA H20 GPUs with 96GB VRAM**. We utilize the DeepSpeed framework with BFloat16 mixed-precision to maximize training throughput and memory efficiency.

### C.2. Algorithm Overview

We present the complete algorithmic framework of ResVLA in Algorithm 1. This pseudocode formally outlines the end-to-end execution flow, detailing the integration of the frequency-domain intent prior (Stage 1) with the residual flow matching refinement (Stage 2) for both training optimization and inference generation.

---

**Algorithm 1** ResVLA: Anchoring Generative VLA Policies with Residual Bridges

---

**Require:** Image $\mathbf{I}$, Language $\mathbf{L}$, Learnable Queries $\mathbf{Q}$, Horizon $T$
**Ensure:** Training Loss $\mathcal{L}$ or Action Trajectory $\mathbf{x}_1$

1: **// Stage 1: Multimodal Encoding & Intent Prior**
2: $\mathcal{C} \leftarrow \text{VLM}_{\text{enc}}(\mathbf{I}, \mathbf{L})$ {Extract multimodal context}
3: $\mathbf{Z} \leftarrow \Phi_{\text{enc}}(\mathbf{Q}, \mathcal{C})$
4: $\mathbf{x}_{\text{anchor}} \leftarrow \Phi_{\text{decoder}}(\mathbf{Z})$
5: **// Stage 2: Residual Flow Matching (Layers $K{+}1{\sim}L$)**
6: **if** Training **then**
7:     Sample $t \sim \mathcal{U}(0,1), \epsilon \sim \mathcal{N}(\mathbf{0}, \sigma_{\min}^2 \mathbf{I})$
8:     $\mathbf{x}_{\text{src}} \leftarrow \mathbf{x}_{\text{anchor}} + \epsilon$ {Source distribution centered at anchor}
9:     $\mathbf{x}_t \leftarrow (1-t)\mathbf{x}_{\text{src}} + t\mathbf{x}_{\text{gt}}; \quad \mathbf{u}_t \leftarrow \mathbf{x}_{\text{gt}} - \mathbf{x}_{\text{src}}$ {Optimal Transport path}
10:     $\hat{\mathbf{v}} \leftarrow \Phi_{\text{FM}}(\mathbf{x}_t, t, \mathcal{C})$ {Predict vector field conditioned on $\mathcal{C}$}
11:     **return** $\mathcal{L} = ||\hat{\mathbf{v}} - \mathbf{u}_t||^2 + \lambda ||\mathbf{x}_{\text{anchor}} - \mathcal{F}_{\text{idct}}^{-1}(\mathcal{P}_\mathcal{S}(\mathcal{F}_{\text{dct}}(\mathbf{x}_{\text{gt}})))||^2$
12: **else**
13:     Sample $\epsilon \sim \mathcal{N}(\mathbf{0}, \sigma_{\min}^2 \mathbf{I})$
14:     $\mathbf{x}_0 \leftarrow \mathbf{x}_{\text{anchor}} + \epsilon$ {Initialize flow from noisy anchor}
15:     **for** $i = 0$ to $N-1$ **do**
16:         $\mathbf{x}_{t+\Delta t} \leftarrow \mathbf{x}_t + \Phi_{\text{FM}}(\mathbf{x}_t, t_i, \mathcal{C}) \cdot \Delta t$ {Euler Integration}
17:     **end for**
18:     **return** $\mathbf{x}_1$
19: **end if**

---

## C.3. Hyperparameters

To facilitate reproducibility, we list the key hyperparameters used for training ResVLA. For model training, we utilize the AdamW optimizer with mixed-precision enabled, configured with $\beta$ parameters of $0.9, 0.95$ and a weight decay of $1 \times 10^{-8}$, conducting the training over $40,000$ global steps which include a $5,000$-step warmup period; the learning rate follows a cosine decay schedule where the base learning rate is set to $2.5 \times 10^{-5}$ while differential learning rates are applied to specific modules, assigning $1.0 \times 10^{-5}$ to the Qwen-VL interface and $1.0 \times 10^{-4}$ to the action model; furthermore, we implement gradient clipping with a norm threshold of $1.0$ and scale the loss weights for the VLA. Unless otherwise noted, all baseline results reported in the main simulation tables are cited from prior work rather than independently reproduced.

