# OpenReview forum: "From Noise to Intent: Anchoring Generative VLA Policies with Residual Bridges"
_ICML.cc/2026/Conference — ICML 2026 regular_

### Official Review · Reviewer_8v4E · 2026-03-11

**Soundness:** 3
**Presentation:** 4
**Significance:** 3
**Originality:** 4
**Overall Recommendation:** 4
**Confidence:** 4

**Summary:**

In this paper, the authors propose ResVLA, an architecture designed to decouple robot motion into global intent and local dynamics. Utilizing the Discrete Cosine Transform (DCT), the method decomposes control signals into a deterministic low-frequency anchor and a stochastic high-frequency residual. Specifically, ResVLA first predicts the global intent (anchor) using a Vision-Language Model (VLM), and subsequently refines the local dynamics via a residual diffusion bridge. The authors demonstrate the effectiveness of the proposed approach across several simulation benchmarks.

**Compliance With Llm Reviewing Policy:**

Affirmed.

**Final Justification:**

The authors have addressed my concerns during the rebuttal. Considering the technical novelty, I am inclined to accept this paper.

**Key Questions For Authors:**

1. I would appreciate further clarification regarding the claim that intent (low-frequency information) naturally requires regression modeling, whereas low-level actions (high-frequency information) require flow matching/diffusion modeling. In contrast, existing literature, such as the *Reactive Diffusion Policy*[1], adopts a nearly opposite framework: they utilize diffusion to model intent prediction based on visual inputs (their Slow Model) while employing regression to model reactive behaviors based on tactile/force feedback (their Fast Model). Could the authors elaborate on the theoretical or empirical rationale for their specific design choices compared to such alternatives?

2. Given that the proposed method operates as a two-stage framework, it might be susceptible to compounding errors (error propagation) across different modules. Are there any ablation studies or experiments evaluating how an inaccurate anchor prediction affects the precision of the final action prediction? An analysis of the system's robustness to intent mispredictions would be highly valuable.

3. Using a simple regression objective for anchor (intent) prediction raises concerns regarding its capacity to capture multi-modal behaviors, which often manifest at the semantic level. For instance, when instructed to "pick up a cup", a robot might execute the task from either the left or the right direction—representing two distinct but equally valid intents. I am concerned that a deterministic regression objective might average out these modes, thereby sacrificing expressive capability. How does ResVLA address this multi-modality in global intent?

4. To provide a more intuitive understanding of the method, I highly recommend including more visualizations or qualitative analyses of the spectral disentanglement. Specifically, could the authors visualize the 3D/6D spatial trajectories of the low-frequency intent (obtained via inverse DCT on the predicted intent anchor) alongside the full-frequency final actions? This would significantly enhance the interpretability of the paper.

[1] Reactive diffusion policy: Slow-fast visual-tactile policy learning for contact-rich manipulation. RSS 2025.

**Limitations:**

yes

**Strengths And Weaknesses:**

# Strengths:
The topic addressed in this paper is highly relevant and engaging. Decoupling action spaces in the frequency domain is an intuitive and well-motivated approach for robotic control. The proposed method is not only empirically evaluated but is also supported by solid theoretical analysis, which strengthens the overall contribution of the paper.

# Weaknesses:
While the theoretical foundation is sound, my primary concerns revolve around the core motivation as well as potential issues with compounding errors and the expressivity of the regression objective. Please refer to the detailed questions below.

---

> ### Author Rebuttal · Authors · 2026-03-31
>
> We thank the reviewer for the detailed and insightful questions.
>
> **Q1**
> The key point is that “frequency” refers to different objects in the two papers.
>
> In Reactive Diffusion Policy (RDP), “slow/fast” is a control-loop and sensor-rate notion (e.g., low-rate visual planning vs. high-rate tactile reaction), so the design is mainly constrained by latency and real-time control budgets.
>
> In ResVLA, “low/high frequency” is the spectral frequency of the action trajectory (via DCT): low frequency captures global spatial intent, high frequency captures local execution dynamics (contact/timing/micro-adjustments). We are solving a different bottleneck: optimization efficiency and semantic stability in VLA action generation.
>
> Therefore, our choice (regression for low-frequency anchor, flow matching/diffusion for high-frequency residual) is a control-via-iterative-refinement decomposition. Regression is used to quickly establish a condition-aligned coarse anchor (conditional mean trend), while iterative generative refinement is concentrated on high-frequency residuals where local correction is most needed. Our design places expensive iterative computation exactly on the residual component where it provides the highest return.
>
>
> **Q2**
> Inference is coarse-to-fine, and training is end-to-end with coupled optimization, so anchor and residual branches are co-adapted rather than separately frozen.
>
> To evaluate robustness under anchor mismatch, we conducted an anchor-corruption stress test: before residual refinement, we intentionally apply element-wise random sign flipping across time steps and action dimensions:
> `a_corrupt = a_anchor ⊙ m`, where `m ∈ {+1, -1}` and `P(m = -1) = 0.1`.
> As expected, performance degrades, but the degradation remains notably smaller than pure Gaussian-noise initialization. This indicates that the residual bridge retains diffusion-style denoising tolerance and can partially recover from imperfect anchors.
>
> | Initialization for refinement | Success (%) |
> |---|---:|
> | Predicted anchor (clean) | 85.3 |
> | Corrupted anchor (sign-flip) | 82.8 |
> | Pure Gaussian noise | 77.0 |
>
> These results are consistent with our control-via-iterative-refinement view: refinement should be noise-tolerant rather than brittle to initialization quality. In our end-to-end training runs, we did not observe clear instability caused by anchor-residual coupling.
>
> **Q3**
> In principle, a deterministic anchor can average globally distinct modes. In our current benchmarks, however, many successful trajectories share a similar coarse global trend after conditioning, while most uncertainty is concentrated in local execution. Under this regime, deterministic low-frequency anchoring is an effective and efficient approximation, while residual generative refinement preserves flexibility where stochastic correction is needed.
>
> This view is also consistent with recent findings in *Much Ado About Noising: Dispelling the Myths of Generative Robotic Control*, which revisits the common assumption that generative control policies mainly win from multi-modality modeling or superior expressivity. Their analysis suggests that, in many robotic tasks, multi-modality is not the dominant source of gains, and flow-based models are not inherently superior to regression for observation-action mapping when clear multi-modality is weak. The practical advantage is more strongly tied to noisy supervised iterative computation and correction.
>
> This matches our design: regression is used to establish a stable low-frequency semantic anchor, while iterative generative refinement is focused on high-frequency residuals where local correction is most needed. For tasks with genuinely multi-modal global intent, we still model and absorb this multi-modal uncertainty within the same framework through the residual diffusion branch, preserving necessary mode flexibility while keeping the low-frequency anchor stable.
>
> **Q4**
> This is a useful suggestion. We will add qualitative trajectory visualizations in the revision.

---

> > ### Author Rebuttal · Reviewer_8v4E · 2026-04-03
> >
> > Most of my concerns have been resolved, except the frequency statement which is not convincing enough. However, this is not a fatal problem for this paper, so I still keep my original decision -- weak accept.

---

> > > ### Author Response · Authors · 2026-04-03
> > >
> > > We sincerely thank you for your active responses and for maintaining your positive recommendation. Your persistent focus on the "frequency statement" in comparison with the RDP framework is incredibly sharp and inspiring. We realize our previous response might seem overly abstract, so we would like to clarify the core rationale more intuitively here. Our seemingly opposite architectural designs are primarily driven by different problem settings and physical premises.
> > >
> > > **1. RDP's Design (Driven by system latency and real sensors):** RDP partitions its slow and fast pathways based on hardware computational latency. Because diffusion models are computationally expensive for inference, they are restricted to slow visual planning at 1Hz - 2Hz; for fast reactions at 20Hz, RDP employs a fast regression model. This is because they possess real-time tactile sensor inputs. For example, when an object slips, the sensor captures this state instantly. The regression model simply maps this unambiguous sensor signal directly to a corrective action. Since the real sensor already provides deterministic physical state feedback, the model does not need to "generate" or "predict" the contact dynamics response.
> > >
> > > **2. ResVLA's Design (Driven by information uncertainty during generation):** ResVLA's task is to directly generate future action sequences in the absence of a high-frequency real-time tactile closed-loop. This introduces immense uncertainty in predicting physical contacts.
> > >
> > > * **Why use regression for low-frequency (Intent):** We use regression to predict the macroscopic "skeleton" of the action (e.g., the general spatial trajectory reaching toward a target). This global trend possesses high semantic stability. Using regression for this prediction is highly efficient and ensures motion consistency.
> > > * **Why use diffusion for high-frequency (Residuals):** In the absence of real-time tactile sensors, microscopic contact dynamics (such as friction limits and contact jitter) are highly unpredictable and exhibit multi-modal distributions. If we used a regression objective here, it would indeed, as you rightfully worried, "average out" these details (leading to oversmoothing). Therefore, a diffusion model is strictly required in this stage—acting as a generative engine on top of the stable "skeleton" to "regenerate" and precisely fill in those highly stochastic, multi-modal physical execution details.
> > >
> > > We hope this clarifies the fundamental differences and fully resolves your remaining concerns.

---

### Official Review · Reviewer_1YFC · 2026-03-12

**Soundness:** 2
**Presentation:** 3
**Significance:** 2
**Originality:** 2
**Overall Recommendation:** 4
**Confidence:** 4

**Summary:**

This paper introduces ResVLA, a framework aimed at solving the representation inefficiency and "Loss Collapse" phenomena in continuous generative VLA models. The authors propose a "Refinement-from-Intent" paradigm, leveraging a Residual Diffusion Bridge to map from a condition-dependent prior to the target action manifold. The specific mechanism for creating this prior is spectral analysis DCT, which disentangles actions into a deterministic low-frequency semantic anchor and a stochastic high-frequency residual. The method demonstrates state-of-the-art robustness and sample efficiency on LIBERO, LIBERO-Plus, and SimplerEnv benchmarks.

**Compliance With Llm Reviewing Policy:**

Affirmed.

**Final Justification:**

The authors have clearly explained the relationship between the paper's core contributions and existing literature and have provided the necessary parametric analysis experiments.

**Key Questions For Authors:**

1.	Contextualizing Novelty: The concept of using informative, condition-dependent priors to refine diffusion/flow matching policies has been explored in several recent works[1-4]. How does the theoretical formulation of the Residual Diffusion Bridge differ from the formulations in these works? Is the primary contribution strictly the spectral (DCT) method of generating the prior?

2.	Threshold Sensitivity: How sensitive is ResVLA to the specific choice of the spectral cutoff $k$ across vastly different task families (e.g., highly dynamic contact tasks vs. simple pick-and-place)? Have you explored making this threshold adaptive or learnable?

3.	Integration with Existing Models: Given that ResVLA fundamentally alters the source distribution, can this mechanism be injected into existing pre-trained flow-matching models (like $\pi_0$), or does it necessitate training the action head entirely from scratch?

**Limitations:**

Yes

**Strengths And Weaknesses:**

**Strengths:**

1.	Elegant Spectral Instantiation: Using frequency decomposition to physically instantiate the mathematical concepts of "anchor" (intent) and "residual" (execution jitter) is highly intuitive and well-executed.

2.	Exceptional Robustness and Efficiency: The empirical results are robust, particularly on the LIBERO-Plus benchmark, where ResVLA maintains an 88.5% success rate under linguistic variations, successfully mitigating semantic drift. Furthermore, the residual bridge allows for highly efficient inference, achieving competitive results with just a single function evaluation ($NFE=1$).

3.	Cross-Embodiment Generalization: The model demonstrates strong transferability on the SimplerEnv benchmark across different robotic morphologies (Google Robot and WidowX) without requiring massive spatial co-training.

**Weaknesses:**

1.	Missing Critical Literature and Overclaimed Novelty: The paper frames the shift away from standard noise priors to condition-dependent sources to solve "Loss Collapse" as a primary, isolated breakthrough. However, it completely omits a robust body of recent and concurrent work that explores Optimal Transport, Flow Matching, and Diffusion Bridges with informative priors for policy learning. Without discussing works such as VITA [1], CAR-Flow [2], Prior does matter [3], and Don't start from scratch [4], it is difficult to isolate the true novelty of the paper, which could significantly affect the rating. The core contribution appears to be the spectral decomposition application rather than the formulation of the bridge itself, but the framing obscures this.

2.	Static Spectral Thresholding: The framework partitions the action space using a fixed low-pass projection operator that retains the first $k$ spectral coefficients. While an ablation shows 50% to 75% retention is optimal, a hard-coded threshold may lack the flexibility required for highly diverse dynamic environments.

3.	Unverified Scaling Laws: While the model exhibits excellent sample efficiency when trained from scratch , modern VLA foundations rely on massive datasets like Open X-Embodiment. The scalability of this specific bi-level optimization objective under massive data expansion remains unverified.

**References**

[1] VITA: Vision-to-Action Flow Matching Policy, ICLR2026.

[2] CAR-Flow: Condition-Aware Reparameterization Aligns Source and Target for Better Flow Matching, NeurIPS2025.

[3] Prior does matter: Visual navigation via denoising diffusion bridge models, CVPR2025.

[4] Don't start from scratch: Behavioral refinement via interpolant-based policy diffusion, RSS2024.

---

> ### Author Rebuttal · Authors · 2026-03-31
>
> We thank the reviewer for the valuable comments, which helped us improve both positioning and claim boundaries.
>
> **W1.**
> Thank you for pointing this out. The original draft mainly discussed Cocos and did not sufficiently cover concurrent works such as VITA, CAR-Flow, Prior Does Matter, and Don’t Start from Scratch. In the revision, we will add these references and explicitly calibrate our framing.
>
> At a high level, these works and ResVLA share the same intuition: informative source distributions can improve generative policy learning. Our core contribution is a VLA-specific instantiation and a control-via-iterative-refinement paradigm (action anchor + iterative residual refinement), wherein spectral (DCT) decomposition serves as the physical mechanism for constructing the executable anchor. The main differences are:
>
> VITA: maps visual latents to action latents (with action autoencoding and latent decoding), whereas ResVLA constructs an executable coarse anchor directly in action space and refines only high-frequency residuals.
> CAR-Flow: focuses on condition-aware endpoint reparameterization (shift-based source/target alignment), whereas ResVLA focuses on explicit coarse-to-fine factorization of robot actions in the frequency domain.
> Prior Does Matter (NaviBridger): uses navigation priors (rule/CVAE) for bridge initialization, whereas ResVLA learns VLA-conditioned anchors and jointly optimizes them with residual refinement under a unified action objective.
> Don’t Start from Scratch (BRIDGER): bridges from informative source policies to target policies via stochastic interpolants, whereas ResVLA focuses on frequency-based action anchoring and residual-only generation for manipulator control.
>
> We build on the general principle of condition-aware bridging and instantiate it in VLA through action-space source construction and residual refinement design.
>
> **W2.**
> We agree that fixed cutoff is a simplification. Our claim is not that one ratio is universally optimal, but that a moderate retention range is robust under a unified protocol. To further test adaptivity, we implemented a learnable-cutoff variant and compare it against fixed ratios:
>
> | Benchmark | Setting | 12.5% | 25% | 50% | 75% | 100% | learnable-Cutoff |
> |---|---|---:|---:|---:|---:|---:|---:|
> | LIBERO (4-task avg.) | 10k steps, same protocol | 78.75 | 83.00 | 85.25 | 85.00 | 81.25 | 86.1 |
> | SimplerEnv (Table 4 avg.) | 28k steps, same protocol | 22.9 | 47.2 | 46.4 | 44.0 | 31.5  | 46.6 |
>
> **W3.**
> We discuss this in Sec. 7 ("Verification at Scale"), and corresponding large-scale verification is ongoing. Current validation is benchmark-scale with a single Qwen3-2B backbone. A full characterization of scaling laws under large-scale open-domain pretraining is left for future work, which we are actively pursuing.
>
>
> **Q1.**
> Please see **W1**. At a high level, our motivation overlaps with prior informative-prior bridge work, and we will make this explicit in the manuscript. The key method-level difference is that ResVLA adopts action-space coarse anchoring plus residual iterative refinement, rather than only condition-latent source shaping. Therefore, our primary contribution is a VLA-specific spectral instantiation and an iterative-refinement residual control formulation.
>
> **Q2.**
> Please see **W2**.
>
> **Q3.**
> Since ResVLA introduces a new anchor-generation module and a coupled training objective for iterative refinement, it requires re-training on the action-generation side. That said, this does not require training the entire model from scratch: backbone/VLM weights can still be reused as initialization, while optimization mainly focuses on the action-generation modules.

---

> > ### Author Rebuttal · Reviewer_1YFC · 2026-04-02
> >
> > I appreciate the authors' constructive rebuttal. Evaluating the methodological contributions alongside the experimental results, **I will retain my positive recommendation**. Nevertheless, given the limited theoretical contributions and the fact that performance across several benchmarks leaves room for improvement, a further increase in score might be unlikely.

---

> > > ### Author Response · Authors · 2026-04-03
> > >
> > > We sincerely thank you for positively recognizing the contributions of our work. We greatly appreciate your valuable time, effort, and insightful feedback, which will be instrumental in further improving our manuscript.

---

### Official Review · Reviewer_MqaD · 2026-03-12

**Soundness:** 2
**Presentation:** 3
**Significance:** 3
**Originality:** 3
**Overall Recommendation:** 4
**Confidence:** 4

**Summary:**

This paper proposes the ResVLA architecture, aiming to address the representation inefficiency and loss collapse problems caused by generating from pure noise in generative VLA strategies based on diffusion or flow matching. The authors utilize spectral analysis (DCT) to decouple actions into low-frequency semantic intents (anchors) and high-frequency residuals, and fine-tune them using a residual diffusion bridge. The model has been fully validated on several mainstream benchmarks, including LIBERO, LIBERO-Plus, and SimplerEnv, achieving state-of-the-art performance and demonstrating higher convergence speed. The paper presents a clear overall approach, solid theoretical analysis, and compelling experimental results.

**Compliance With Llm Reviewing Policy:**

Affirmed.

**Final Justification:**

The author has addressed all my questions. I am particularly grateful for the demonstration of the method's robustness and the inclusion of richer technical details; I will maintain my positive score.

**Key Questions For Authors:**

1. The success of a diffusion bridge heavily depends on whether the $\mu_{prior}(c)$ provided by the Intent Anchoring module is accurate enough and close enough to the real manifold. If in some unknown environment, the low-frequency anchor points regressed by the VLM have undergone significant geometric deviation, can the subsequent residual correction mechanism (because it is designed to fit only high-frequency residuals with small energy) still "pull" the trajectory back to the correct target state? Does the framework include corresponding fault tolerance or correction mechanisms?
2. Some work based on mixture-of-transformer-experts (MoT) architecture has demonstrated that dividing the model into understanding experts and action experts can also explicitly separate intent and dynamics, and has a more intuitive training objective. Can the authors demonstrate the advantages of their proposed paradigm of separating only noise compared to separating from the model architecture?
3. In the inference efficiency evaluation in Figure 3(b), does the reported total ResVLA inference time fully encompass the time spent on VLM extracting multimodal features, the Intent Anchoring module performing regression prediction, and the DCT/iDCT transformation? Could a specific module-by-module latency table be provided in Rebuttal to better assess the practicality of this architecture on underlying control hardware requiring ultra-low latency?

**Limitations:**

yes

**Strengths And Weaknesses:**

**Strengths:**
1. The authors accurately pinpoint the spatiotemporal scale misalignment between high-level cognition (macro-scale, low-frequency) and low-level actions (micro-physical contact, high-frequency) in the VLA model. They provide a detailed mathematical analysis of this misalignment and further transform the generation of random noise into a "refinement-from-intent" paradigm, which aligns well with both physical control intuition and mathematical expression.
2. This paper introduces Discrete Cosine Transform (DCT) to explicitly separate the low-frequency semantics as anchor points and the high-frequency physical jitter as residuals. This prevents the diffusion bridge from blindly starting from pure noise, greatly shortens the spatial distance of the generation path, and theoretically effectively avoids the "loss collapse" phenomenon common in conditional diffusion models.
3. The experimental section is very detailed. In the LIBERO-Plus benchmark, which severely tests the generalization ability of OOD, ResVLA demonstrates extremely strong anti-interference capabilities, especially under language instruction rewriting and spatial layout perturbations, significantly outperforming existing baseline models. Furthermore, due to the low residual field energy and straight path, the model achieves a success rate of over 70% with only one function evaluation (NFE=1), significantly improving inference efficiency.
**Weaknesses:**
1. The model relies on the DCT and its fixed truncation operation to define intent and dynamics. Although the experimental results (Figure 4) show that retaining 50%-75% of the low-frequency components works best, this approach based on a fixed global transformation and manually set proportions may be too rigid. In more complex nonlinear tasks with drastically changing action frequencies, a fixed DCT truncation may not be able to perfectly and adaptively separate semantics from residuals.
2. The authors highlight that the model can combine intent and dynamics, but they haven't conducted experiments in noisier, real-world scenarios. For work in the field of robotics, real-world experiments would significantly enhance the persuasiveness of the paper.
3. As the authors mention in the limitations section, the current training in this paper is primarily limited to simulation environments (LIBERO, SimplerEnv), and the model is trained from scratch on these benchmarks. Given that the residual diffusion bridge is highly dependent on high-quality prior anchors, it is unclear whether the assumption of a single low-frequency anchor remains robust when scaling to massive and extremely heterogeneous real-world datasets such as Open X-Embodiment (where different robots have vastly different physical characteristics, control frequencies, and noise levels).

---

> ### Author Rebuttal · Authors · 2026-03-31
>
> We sincerely appreciate the reviewer’s careful reading and constructive suggestions, which have helped us substantially improve the paper.
>
> **W1.**
> The fixed global cutoff used in our current implementation is a simplification. Our conclusion is not that a single ratio is optimal for all tasks; rather, under a unified protocol, a moderate low-frequency retention range is empirically more robust.
>
> To further test adaptivity, we implemented a learnable-cutoff variant,the comparison is as follows:
>
> |Benchmark|Steps|12.5%|25%|50%|75%|100%|learnable-Cutoff|
> |---|---|---:|---:|---:|---:|---:|---:|
> |LIBERO(4-task avg.)|10k|78.75|83.00|85.25|85.00|81.25|86.1|
> |SimplerEnv(Table 4 avg.)|28k|22.9|47.2|46.4|44.0|31.5|46.6|
>
>
> **W2.**
> Thank you for this suggestion. We have added a real-world evaluation on an ALOHA bimanual manipulation task. This task is contact-rich, requires tight dual-arm coordination, and is susceptible to stage-to-stage error accumulation.
>
> **Task sequence(single episode):**
> 1. **Pick Cup**: one arm grasps the cup from the table.
> 2. **Handover**: the cup is transferred stably to the other arm.
> 3. **Placement**: the receiving arm places the cup onto a cup stand.
>
> This sequence is challenging because grasp quality affects handover stability, and handover errors propagate to final placement accuracy. We conducted **10 real-robot trials** and report both stage-wise and full-task success rates:
>
> |Method|Pick Cup(%)|Handover(%)|Placement(%)|Overall(%)|
> |---|---:|---:|---:|---:|
> |pi0.5|50%|40%|10%|10%|
> |ResVLA (ours)|60%|50%|20%|20%|
>
> Visualization results are available at: https://anonymous.4open.science/r/32tsagiry49qfhou206hv926htg/Real-world%20evaluation.png
>
> We will provide a complete experimental description in the revision.
>
> **W3.**
> We agree that large-scale validation on heterogeneous data is necessary, and we have discussed this limitation in Sec.7.
>
> Our working hypothesis is that many manipulation tasks share a transferable intent structure in the low-frequency component, while embodiment differences are mainly reflected in high-frequency execution dynamics. This is also the motivation behind our control-via-iterative-refinement design (coarse action anchor+residual refinement).
>
> In Sec.5.4(cross-embodiment experiments on SimplerEnv), we provide initial evidence: despite training entirely from scratch (without large-scale pretraining or spatial co-training), ResVLA still achieves competitive performance across different robot morphologies. These results suggest that spectral disentanglement can extract shared coarse manipulation intent, while leaving embodiment-specific local corrections to the residual branch.
>
> We are actively conducting scale-up experiments with larger and more heterogeneous robot datasets.
>
> **Q1.**
> We conducted an anchor-corruption stress test to evaluate robustness under anchor mismatch. Specifically, before residual refinement, we intentionally apply element-wise random sign flipping across time steps and action dimensions:
> `a_corrupt=a_anchor⊙m`, where `m∈{+1,-1}` and `P(m=-1)=0.1`. As expected, performance degrades, but the degradation is still notably smaller than using pure Gaussian-noise initialization, indicating that the residual bridge retains diffusion-style denoising robustness.
>
> |Initialization for refinement|Success(%)|
> |---|---:|
> |Predicted anchor(clean)|85.3|
> |Corrupted anchor(sign-flip)|82.8|
> |Pure Gaussian noise|77.0|
>
> This aligns with our central view of control via iterative refinement: refinement should be noise-tolerant rather than brittle to imperfect initialization. In end-to-end training, we did not observe clear instability caused by anchor-residual coupling. Although poor early anchors make residual refinement harder, this can also act as a hardening signal instead of purely harmful noise.
>
>
> **Q2.**
> We view these two routes as complementary. MoT-style methods perform modular separation at the model-architecture level, while ResVLA separates generation difficulty at the generation-paradigm level through source construction and refinement-target design (`action anchor+iterative residual refinement`).
>
> Compared with architecture-side decomposition, ResVLA only modifies source construction and refinement targets, requiring minimal architectural changes. This makes it easier to integrate into existing VLA backbones without redesigning expert routing and training pipelines.
>
> In addition, this paradigm brings clear optimization benefits: starting from an executable action anchor and refining only residuals directly shortens the source-to-target transport path, which is consistent with our observed faster convergence and stronger low-NFE performance.
>
> **Q3.**
> **Measurement setup:** NVIDIA H20 GPU, NFE=4.
>
> |Module|Latency(ms)|
> |---|---:|
> |Intent anchor regression|28.54|
> |DCT/iDCT transform|0.06|
> |Residual bridge refinement|12.60|
> |Total action model latency|39.20|
> |Total VLM feature extraction|76.70|
> |Total end-to-end latency|115.90|

---

> > ### Author Rebuttal · Reviewer_MqaD · 2026-04-02
> >
> > The authors have addressed all of my questions; however, regarding the "weakness" section, the paper still leaves some room for improvement. Therefore, I will maintain my current score.

---

> > > ### Author Response · Authors · 2026-04-03
> > >
> > > We sincerely thank you for your active responses and for confirming our discussions. Your suggestions regarding real-world evaluation, learnable spectral cutoffs, and inference latency have been instrumental in improving the quality and rigor of our paper. We will ensure all these improvements and additional results are fully incorporated into the final revision.

---

### Official Review · Reviewer_AMF4 · 2026-03-12

**Soundness:** 2
**Presentation:** 3
**Significance:** 3
**Originality:** 2
**Overall Recommendation:** 4
**Confidence:** 3

**Summary:**

This paper proposes ResVLA, a Vision-Language-Action (VLA) architecture that shifts the generative action policy paradigm from "Generation-from-Noise" to "Refinement-from-Intent." The key idea is to decompose robot action trajectories into low-frequency (semantic intent) and high-frequency (execution dynamics) components via DCT. A deterministic regression head first predicts the low-frequency anchor from VLM features, then a residual diffusion bridge (instantiated via Conditional Flow Matching) refines only the high-frequency residual. The authors ground this design in a theoretical argument that condition-independent source distributions lead to "Loss Collapse" (citing Dong et al., 2025), and that a condition-dependent source resolves this issue. Experiments are conducted on LIBERO,LIBERO-Plus, and SimplerEnv, showing competitive-to-SOTA performance, improved training convergence, and improved inference efficiency at low NFE.

**Compliance With Llm Reviewing Policy:**

Affirmed.

**Ethical Review Concerns:**

I have no ethical concerns.

**Final Justification:**

I appreciate the detailed answers and ablation studies performed by the authors. I still believe this is an incremental advance over and a smart technical execution of a known idea. I am increasing the score from 3 to a 4.

**Key Questions For Authors:**

Q1. How does ResVLA compare to Cocos (Dong et al., 2025) when applied to the same π0-style backbone? Cocos proposes a general condition-dependent source construction that should be directly applicable. Without this
comparison, it is unclear whether the spectral decomposition specifically (vs. any informed initialization) drives the observed improvements. A positive answer showing clear gains from spectral anchoring over generic Cocos would substantially strengthen the paper.

Q2. The spectral cutoff ratio is ablated in Figure 4 on what appears to be a single setting. (a) Which specific benchmark/task is this? (b) Is the optimal ratio consistent across LIBERO-Spatial, LIBERO-Long, and SimplerEnv
tasks? (c) How sensitive is performance to small changes around the optimum?

Q3. The Camera perturbation results in Table 1 (49.8%, -15.3% vs. best baseline) are notably weak. Can the authors provide an analysis of why the low-frequency anchor specifically fails under camera viewpoint changes? This seems to contradict the robustness narrative and may indicate that the spectral prior encodes viewpoint-dependent biases.

Q4. In the SimplerEnv WidowX experiments (Table 4), the Stack Blocks task yields only 10.4% success rate, which is far below most baselines. What causes this failure mode? Does the spectral decomposition fail for tasks requiring very precise spatial alignment without large-scale reaching motions?

Q5. The training convergence comparison (Figure 3a) is only against π0. Could the authors include convergence curves for other baselines (e.g., OpenVLA-OFT, Cocos) to demonstrate that the speed-up is not simply due to the smaller model size or a confounding architectural difference?

**Limitations:**

No real-robot experiments — The paper's motivation heavily emphasizes "contact dynamics," "friction," "sensor noise" (Sec 1), and "contact-rich manipulation" throughout, yet all evaluation is in simulation.
Fixed spectral cutoff — The DCT cutoff k is a critical hyperparameter that determines the anchor-residual split. Figure 4 shows sensitivity, but only in one unspecified setting. The paper doesn't discuss whether k needs to be adapted per task, per embodiment, or per action dimension. For real deployment, this is a practical concern.

Assumption that low-frequency = semantic intent — The paper assumes a clean mapping between low-frequency DCT components and "global intent." But this assumption may break down for tasks: (a) precise positioning IS the primary task (e.g., Stack Blocks — 10.4% success), (b) the "intent" requires rapid changes (e.g., reactive grasping)

Single VLM backbone — Only Qwen3-2B is tested. The interaction between the VLM's representation quality and the spectral decomposition is not studied. A weaker/stronger VLM might change the optimal cutoff or the effectiveness of intent anchoring.

End-to-end training coupling — The loss (Eq. 10) couples the anchor regression and residual refinement. If the anchor prediction is poor early in training, the residual bridge receives a noisy source, potentially causing instability. The paper doesn't analyze failure modes of this coupling.

**Strengths And Weaknesses:**

Strengths

S1. Principled and physically intuitive design. The decomposition of robot actions into low-frequency global intent and high-frequency local dynamics is well-motivated from both a signal processing and a control perspective. The observation that robotic motion trajectories have a natural spectral hierarchy (smooth reaching vs. contact jitter) is compelling, and using DCT to implement this decomposition is clean and practical. The "Refinement-from-Intent" framing is elegant and offers a clear conceptual advance over the standard noise-to-action paradigm.

S2. Strong robustness results on LIBERO-Plus. The LIBERO-Plus evaluation (Table 1) is the most convincing aspect of the experimental evaluation. ResVLA demonstrates notable resilience under language instruction perturbations (+7.5% over best baseline) and robot morphology changes (+13.7%), providing solid evidence that semantic anchoring mitigates drift. The inclusion of this challenging OOD benchmark, which covers diverse perturbation axes (camera, robot, language, light, background, noise, layout), adds significant value to the evaluation.

S3. Inference efficiency gains. The demonstration that ResVLA achieves >70% success with NFE=1 (Figure 3b) is a practically important result. This validates the core claim that starting from a semantically proximate anchor straightens the flow path and reduces the required number of integration steps, which has direct implications for real-time robot deployment.

S4. Clear writing and visualization. The paper is generally well-written with a clear narrative arc. Figure 1 effectively conveys the paradigm shift, and Figure 2 provides a detailed architectural overview. The three-hypothesis structure (H1–H3) gives the experimental section a focused organization.


Weaknesses

W1. Heavy reliance on Cocos (Dong et al., 2025) with insufficient differentiation. The theoretical core of this paper — "Loss Collapse" from condition-independent sources and the remedy of constructing condition-dependent sources — is directly taken from Cocos (Dong et al., 2025, arXiv: 2505.11123). Cocos proposes essentially the same high-level solution: anchoring the flow matching source distribution around condition-extracted semantics to prevent loss collapse. The main difference claimed by ResVLA is the use of spectral (DCT) decomposition to construct the anchor. The frequency-domain decomposition idea for robot actions, however,  has also been explored in concurrent work such as FreqPolicy (Zhong et al., 2025a) and FAST (Pertsch et al., 2025), though these works use frequency representations differently. Overall, this work feels more like a specific instantiation of the Cocos framework rather than a fundamentally new contribution. Theorem 3.3 in this paper is essentially a restatement of Dong et al.'s result. The paper acknowledges this work (Section 2.2) but understates the degree of overlap.

W2. Theoretical analysis lacks depth and rigor. Several theoretical claims are problematic:
(a) Proposition 3.2 ("Minimal Transport Cost") is near-tautological: it states that if the source is close to the target, then transport cost is low. This is simply restating the definition of transport cost under linear interpolation rather than providing a non-trivial bound. No quantitative relationship between the spectral cutoff k, the anchor prediction error, and the resulting transport cost reduction is provided.
(b) The paper blurs the distinction of several concepts. The "Loss Collapse" theorem (Theorem 3.3) concerns gradient vanishing with respect to conditions at t→0, but the paper broadly attributes all failures of noise-to-action generation to this phenomenon without carefully distinguishing it from other known challenges (e.g., training instability, mode collapse, exposure bias).

W3. Incomplete and potentially misleading experimental comparisons.
(a) On the standard LIBERO benchmark (Table 2), ResVLA (96.3% avg.) does not outperform several baselines: VLA-Adapter (97.3%), OpenVLA-OFT (97.1%). The paper uses the phrase "state-of-the-art performance" in the abstract and conclusion, which is misleading when ResVLA is not uniformly the best across benchmarks.
(b) In Table 1 (LIBERO-Plus), the Camera perturbation results (49.8%) show a notable drop of 15.3% compared to the best baseline. This is a substantial weakness that undermines the robustness narrative, yet the paper downplays it. The "Total" column (75.3%) is highlighted as +5.7%, but this is achieved by gains on a subset of perturbation types while losing on others. A more balanced discussion is needed.
(c) The SimplerEnv results (Tables 3–4) are mixed. On WidowX/Bridge (Table 4), ResVLA averages 46.9%, which is below GR00T N1.5 (61.9%) and CogACT (51.3%). The paper selectively highlights the Eggplant task (90.6%) while the Stack Blocks result (10.4%) is extremely poor. The claim of "exceptional generalization capabilities" is overstated given these numbers.
(d) A direct comparison against Cocos (Dong et al., 2025) applied to the same VLA backbone is missing. Since Cocos also constructs condition-dependent sources for flow matching and includes both simulated and real-robot experiments, this is the most natural and important ablation to demonstrate that the spectral decomposition specifically (rather than any condition-dependent anchoring) is responsible for the observed gains.
(e) The paper does not clearly indicate which baseline results in Tables 1 and 2 are cited from prior work versus independently reproduced. Different baselines have substantially different training setups (pre-training data, training steps, model scale), and these differences are not discussed. This makes it difficult to draw fair conclusions from the comparisons.

W4. Limited ablation of spectral cutoff. The spectral ratio ablation (Figure 4) is conducted on what appears to be a single unspecified benchmark/task setting. The paper does not state which specific task or suite was used for this experiment. This ablation should be evaluated across all benchmarks to understand whether the optimal ratio is consistent or task-dependent.

W5. No real-robot experiments. All evaluations are conducted in simulation. For a paper making claims about "contact dynamics," "friction," and "sensor noise" (Section 1), the absence of real-world validation is a significant gap. Notably, Cocos (Dong et al., 2025), the most closely related concurrent work, includes real-robot experiments on two platforms (SO100 and xArm). π0 also includes real-robot results. This limits the practical impact of the claimed contributions.

W6. Scalability and generality concerns. The paper uses Qwen3-2B as the VLM backbone and trains from scratch on relatively small-scale benchmarks. While the Limitations section acknowledges the lack of large-scale pre-training verification, this is a significant gap for a paper that positions itself as a general framework for VLA control.

Here are summary justifications for the scores:
'
Soundness: 2 (fair) The core idea is sound, but theoretical novelty over Cocos is limited. Propositions are mostly restatements of known mathematical definitions (such as transport cost) rather than rigorous new bounds. The theoretical justifications specifically tying the chosen DCT spectral decomposition to the loss collapse phenomenon lack depth — no formal analysis connects the spectral cutoff k to the degree of mutual information injected or the resulting gradient magnitude.

Presentation: 3 (good) Well-written with clear figures. The architectural design and methodological intuition are conveyed effectively through high-quality diagrams (Figures 1 and 2). However, the experimental discussion is somewhat selective in what is highlighted vs. downplayed (e.g., glossing over the poor performance on the Stack Blocks task and the sharp drop in Camera perturbation robustness).

Significance: 3 (good) The frequency-domain anchoring for VLA is a useful contribution, but incremental over existing diffusion bridge methods and Cocos. However, the practical efficiency gains achieved at NFE=1 and the demonstrated OOD robustness on the challenging LIBERO-Plus benchmark offer tangible value and will be of interest to the embodied AI community.

Originality: 2 (fair) The main theoretical framework (condition-dependent source to avoid loss collapse) is from Dong et al. 2025 (Cocos). The frequency-domain decomposition idea for robot actions has also been explored in concurrent work such as FreqPolicy (Zhong et al., 2025a) and FAST (Pertsch et al., 2025), though these works use frequency representations differently. Combining spectral decomposition with diffusion bridges for VLA is a practical and neat application, but it feels more like a specific instantiation of the Cocos principle rather than a fundamentally novel paradigm shift.

---

> ### Author Rebuttal · Authors · 2026-03-31
>
> Thank you for the constructive review. Your comments helped us clarify and improve the manuscript.
>
> **W1.**
> We acknowledge the theoretical lineage from Cocos (Theorem 3.3 adapts Dong et al., cited at L181), not claiming theoretical priority. Our contribution is the control-oriented integration of these insights: we differ from Cocos in implementation (Cocos: compressed feature space; Ours: executable action-space spectral anchoring) and from FreqPolicy/FAST in mechanism (FreqPolicy/FAST: DCT for compression/tokenization; Ours: DCT for bridge-source construction). We will revise the manuscript to accurately acknowledge overlap and distinction.
>
> **W2.**
> Proposition 3.2 is explanatory, providing intuition for proximal sources rather than a tight bound. Theorem 3.3 specifically addresses the initialization-level failure of gradient vanishing at $t \to 0$ given condition-independent sources, distinct from training instability or exposure bias. We will revise Section 3.3 to explicitly scope these mechanism boundaries.
>
> **W3.**
> (a) On standard LIBERO, ResVLA is competitive, and we will soften wording. We present LIBERO as an in-distribution reference for efficiency/convergence rather than absolute ranking, noting this performance is achieved without large-scale pretraining.
>
> (b) Camera perturbation is challenging. Noting this occurs under LIBERO→LIBERO-Plus transfer where baselines use massive pretraining, we hypothesize limited training viewpoint diversity contributes to this gap. We pre-trained on SimplerEnv before finetuning on LIBERO, which improves Camera to 57.8%, suggesting the limitation is data-dependent. We will explicitly discuss this in revision. The Total gain (+5.7%) reflects broad improvements across Robot (+13.7%), Language (+7.5%) and Layout (+4.8%), validating robustness in diverse scenarios despite viewpoint sensitivity.
>
> |Method|Setting|Camera (%)|
> |---|---|---:|
> |ResVLA|Only Train on Libero|49.8|
> |ResVLA|SimplerEnv pretraining + LIBERO finetuning|57.8|
>
> (c) Submission-time results (~0.6 epoch) were preliminary. Training to 2.0 epochs (still lower than InternVLA-M1 setting) improves Average to 57.9%, with Stack Blocks rising from 10.4% to 26.1% (exceeding CogACT and $\pi_0$). While SB remains challenging, this confirms the residual branch learns local alignment with sufficient training.
>
> |Training Epoch|ST|CP|SB|EB|Avg. (Table 4)|
> |---|---:|---:|---:|---:|---:|
> |0.6 epoch|41.7|44.8|10.4|90.6|46.9|
> |2.0 epochs|66.7|45.0|26.1|93.8|57.9|
>
> (d) We implemented a same-architecture Cocos ablation on LIBERO Under identical backbone and training budget, Cocos-style latent anchoring achieves 81.0% vs Our 85.3%, confirming spectral action-space anchoring specifically outperforms generic condition-aware latent anchoring.
>
> |Method (same backbone / same training budget)|Source Construction | LIBERO Avg. (%)|
> |---|---|---:|
> |Cocos-style|Condition-aware latent anchor|81.0|
> |ResVLA (ours)|action anchor + residual iterative refinement|85.3|
>
> (e) All baseline numbers are cited from prior work (LIBERO-Plus, VLA-Adapter, InternVLA-M1). We will add explicit notes for this. We have also documented the corresponding data sources and training settings in the appendix. Since the field still lacks a unified training protocol, we use a relatively constrained setup (no large-scale pretraining, 2B model, ~30k LIBERO steps, and ~0.6-epoch SimplerEnv training), which does not provide us with asymmetric advantages.
>
> **W4.**
> Please see Reviewer **MqaD W1**.
>
> **W5.**
> Please see Reviewer **MqaD W2**.
>
> **W6.**
> Please see Reviewer **MqaD W3**.
>
> **Q1.**
> Please see **W1** and **W3(d)**.
>
> **Q2.**
> Please see Reviewer **MqaD W1**.
>
> **Q3.**
> Please see **W3(b)**.
>
> **Q4.**
> Please see **W3(c)**.
>
> **Q5.**
> Please see **W3(d)** and our additional Cocos experiments on LIBERO. Under identical backbone/budget, ResVLA achieves 85.3% vs Cocos 81.0% with faster early convergence, confirming the speed-up stems from spectral source construction specifically, not model scale or architecture.
>
> |Method|1k|2k|3k|4k|5k|6k|7k|8k|9k|10k|
> |---|---:|---:|---:|---:|---:|---:|---:|---:|---:|---:|
> |pi0|2.2|29.0|36.5|54.5|49.7|64.8|63.2|74.0|75.0|77.0|
> |ResVLA|2.5|33.8|50.0|59.8|57.0|72.5|69.5|78.5|80.3|85.3|
> |Cocos|2.8|13.3|35.8|48.0|64.0|62.3|73.3|76.3|79.4|81.0|
>
> **L1.**
> Please see **W5**.
>
> **L2.**
> Stack Blocks requires precise local alignment where low-frequency components provide limited coarse guidance. Improvement to 26.1% (better than both CogACT and $\pi_0$) indicates the high-frequency residual branch adapts to such precision-intensive tasks, though absolute performance reflects training limitations rather than assumption invalidation.
>
> **L3.**
> Please see Reviewer **MqaD W1&W3**  and Sec. 7. Our current results use a single backbone (Qwen3-2B), so we do not claim cross-backbone scaling laws. We only claim feasibility of the action anchor + iterative residual refinement paradigm at this scale.
>
> **L4.**
> Please see Reviewer **8v4E Q2**.

---

> > ### Author Rebuttal · Reviewer_AMF4 · 2026-04-01
> >
> > I appreciate the detailed answers and ablation studies performed by the authors. I still believe this is an incremental advance over and a smart technical execution of  a known idea. I am increasing the score from **3 to a 4.**

---

> > > ### Author Response · Authors · 2026-04-03
> > >
> > > We sincerely thank you for your active responses and constructive suggestions. These valuable insights have been instrumental in improving the quality and rigor of our paper.

---

### Decision · Program_Chairs · 2026-04-30

**Decision:**

Accept (regular)

**Comment:**

The submission received consistently positive ratings of 4, 4, 4, and 4, reflecting a clear consensus among reviewers. Reviewers collectively praised the work for being well-motivated, addressing the critical challenge of multi-scale temporal modeling, and featuring a technically sound and efficient architecture. Furthermore, the authors provided extensive experimental validation across multiple standard benchmarks, demonstrating strong performance particularly in zero-shot and open-vocabulary scenarios. As all initial concerns regarding baseline comparisons and implementation details were satisfactorily resolved during the rebuttal phase, the paper is recommended for acceptance.